# Raman image-activated cell sorting

Nao Nitta et al.#

The advent of image-activated cell sorting and imaging-based cell picking has advanced our knowledge and exploitation of biological systems in the last decade. Unfortunately, they generally rely on fluorescent labeling for cellular phenotyping, an indirect measure of the molecular landscape in the cell, which has critical limitations. Here we demonstrate Raman image-activated cell sorting by directly probing chemically specific intracellular molecular vibrations via ultrafast multicolor stimulated Raman scattering (SRS) microscopy for cellular phenotyping. Specifically, the technology enables real-time SRS-image-based sorting of single live cells with a throughput of up to ~100 events per second without the need for fluorescent labeling. To show the broad utility of the technology, we show its applicability to diverse cell types and sizes. The technology is highly versatile and holds promise for numerous applications that are previously difficult or undesirable with fluorescence-based technologies.

#A list of authors and their affiliations appears at the end of the paper.

The advent of image-activated cell sorting[1–3] and imaging-based cell picking[4–7] has advanced our knowledge and exploitation of biological systems in the last decade. These foundational technologies mediate information flow between population-level analysis (flow cytometry), cell-level analysis (microscopy), and gene-level analysis (sequencing), making it possible to study, elucidate, and exploit the relations between cellular heterogeneity, phenotype, and genotype[1–7]. Specifically, different from traditional high-content screening[8], their ability to physically isolate target cells from large heterogeneous populations serves as a tool to identify the links between the spatial architecture of molecules within the cell (e.g., protein localization, receptor clustering, nuclear shape, cytoskeleton structure, and cell clustering) and the physiological function of the cell (e.g., proliferation, metabolism, secretion, differentiation, signaling, metastasis, and immune synapse formation) as well as for downstream characterizations (e.g., RNA sequencing and electron microscopy) and applications (e.g., cloning, directed molecular evolution, and selective breeding). For example, Nitta et al. have demonstrated image-activated sorting and cultivation of rare algal mutants with aberrantly distributed carbon-concentrating proteins for highly efficient algal photosynthesis research[1]. Lohr et al. have used an imaging-based cell picking approach for whole-exome sequencing of circulating tumor cells (CTCs) as a tool for CTC genomics[6]. Piatkevich et al. have shown directed evolution of genetically encoded fluorescent voltage reporters by using an imaging-based cell picker with a robotic arm for optogenetic research[7].

Unfortunately, these technologies predominantly rely on fluorescent labeling for cellular phenotyping, an indirect measure of molecules in the cell, which has several critical limitations. First, immunofluorescent staining cannot be used for labeling intracellular molecules in live cells due to the low permeability of cellular membranes[9]. Second, while fluorescent fusion proteins allow for highly specific labeling of intracellular proteins, they require gene transfection, which is not available to all types of cells and proteins[10]. Third, without the molecular specificity provided by these labeling methods, intracellular fluorescent staining with fluorescent dyes suffers from non-specific binding and low quantitative performance since it relies on the difference in their chemical affinity to target biomolecules[11]. Fourth, fluorescent probes are typically bulky and often perturb the function of small biomolecules including metabolites[9]. Finally, fluorescent labeling can lead to immunogenicity and introduction of xenobiotic compounds, such that human cells including human induced pluripotent stem cells (hiPSCs)[12] and chimeric antigen receptor T (CAR-T) cells[13] cannot be fluorescently labeled before in vivo use as cell therapies[14]. To overcome these limitations of fluorescent labeling, in silico labeling based on machine learning of numerous unlabeled (e.g., bright-field, phase-contrast) images has recently been shown as an alternative approach to identifying cellular state, interaction, and drug susceptibility[1,2,15–18], but its application range is constrained by the lack of molecular specificity. We anticipate that the ability to sort cells based on unlabeled yet molecularly specific images with high throughput will significantly extend the utility and applicability of image-activated cell sorting and is, hence, expected to further advance single-cell biology and applications.

In this Article, we demonstrate Raman image-activated cell sorting by directly measuring chemically specific intracellular molecular vibrations via coherent Raman scattering for cellular phenotyping without the need for fluorescent labeling. Previous work has shown the ability to sort cells based on fluorescence images[1–3] or acquire Raman images in continuous flow[19], but has not shown the ability to acquire and process Raman images rapidly enough to sort cells. This requires significant innovations in Raman image acquisition, digital image processing, and seamless integration of them with fluidic and mechanical devices into a complete system, which we demonstrate here to achieve not only Raman image-activated cell sorting, but also at a throughput of ~100 events per second (eps). By virtue of its coherent light-matter interaction and parallel photodetection and real-time digital processing scheme, our Raman microscope based on stimulated Raman scattering (SRS) achieves several orders of magnitude faster Raman signal acquisition than spontaneous Raman scattering microscopy. As a result, our Raman image-activated cell sorter (RIACS) offers ~100 times higher sorting throughput as well as spatial resolution by providing Raman images in comparison to (non-imaging) Raman-activated cell sorting[20–23] that provides only one-dimensional (1D) Raman signal intensities with a moderate throughput of ~1 eps. Compared to conventional Raman microscopy with manual pipetting (which is highly labor-intensive and time-consuming), the RIACS shortens the duration of the entire detection-to-isolation process by a few orders of magnitude (i.e., from years to days, from days to minutes), helping researchers significantly accelerate their discovery-making workflow. Furthermore, to show the broad utility of the RIACS, we demonstrate its applicability to diverse cell types and sizes. The RIACS is a highly versatile platform technology and holds promise for numerous applications that are previously difficult or undesirable with fluorescence-based single-cell sorting or isolation technologies[24].

## Results

**Schematic and function of Raman image-activated cell sorting**. The RIACS is schematically shown in Fig. 1 (Supplementary Fig. 1a–1f for pictures of the primary components of the RIACS). Employing the architecture of the fluorescence-image-activated cell sorter[1,2] as a basis, the RIACS seamlessly integrates (1) a two-dimensional (2D) on-chip hydrodynamic cell focuser, (2) a 3D on-chip acoustic cell focuser for focusing cells into a single stream (Supplementary Fig. 2a–2d, see "cell focusing" in the Methods section for details)[25], (3) an event detector for detecting events (e.g., single cells, debris) and triggering image acquisition (see "event detector" in the Methods section for details), (4) an ultrafast multicolor SRS microscope in a line-focusing geometry for continuous, high-speed, blur-free, sensitive molecular-vibrational imaging of flowing cells (Fig. 1b, c, Supplementary Fig. 3a–3b, see "ultrafast multicolor SRS microscope", "optics-microfluidics integration unit", and "design and fabrication of the microfluidic chip" in the Methods section for details), (5) a real-time Raman image processor composed of multiple field-programmable gate arrays (FPGAs)[26], central processing units (CPUs), and a network switch, all on a 10-Gbps all-IP network[27] for high-speed digital image processing and decision making (Supplementary Fig. 4, see "architecture of the real-time Raman image processor" in the Methods section for details), and (6) an on-chip dual-membrane push-pull cell sorter[28] for rapidly isolating target cells from the cell stream (Supplementary Fig. 2a, 2b, see "on-chip dual-membrane push-pull cell sorter" in the Methods section for details). The basic operation of the RIACS is as follows: suspended cells in a sample tube injected into the RIACS are focused by the hydrodynamic and acoustic focusers into a single stream, detected by the event detector, imaged by the SRS microscope, analyzed by the real-time Raman image processor, and sorted into a collection or waste tube by the dual-membrane push-pull cell sorter triggered by decision signals generated by the image processor. The entire process is operated in a fully automated and real-time manner.

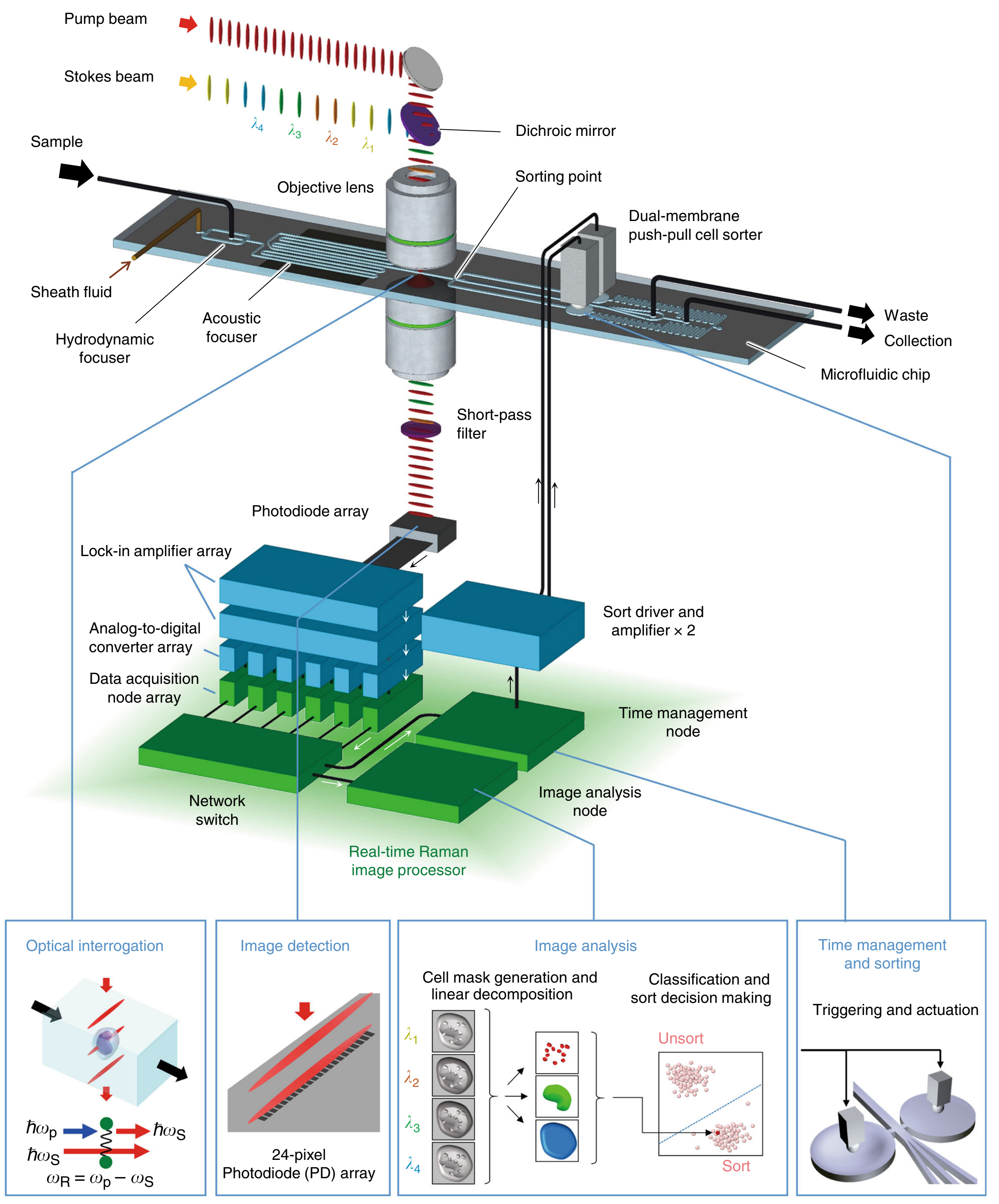


**Fig. 1 Schematic of the RIACS.** Suspended cells injected into the RIACS are focused by the hydrodynamic and acoustic focusers into a single stream, detected by the event detector, imaged by the ultrafast multicolor SRS microscope, analyzed by the real-time Raman image processor, and sorted by the dual-membrane push-pull cell sorter triggered by decisions made by the real-time Raman image processor composed of multiple FPGAs, CPUs, and a network switch, all on a 10-Gbps all-IP network for high-speed digital image processing and decision making. The entire process is operated in a fully automated and real-time manner. Supplementary Fig. 1 shows pictures of the major components of the RIACS.

**Ultrafast multicolor stimulated Raman scattering microscope.** The heart of the RIACS is the ultrafast multicolor SRS microscope, which can acquire four-color, sensitive, blur-free SRS images of cells in a high-speed flow (Supplementary Fig. 3a–3d, see "ultrafast multicolor SRS microscope" in the Methods section for details), which has been impossible with previous SRS microscopes[29–32]. The key elements of the SRS microscope are the high-power pulse-pair-resolved wavelength-switchable laser

with a pulse repetition rate of 38 MHz for time-domain multiplex detection of four-color SRS signals and the line-focusing optics and parallel photodetection system for space-domain multiplex detection of the SRS signals (see "pulse-pair-resolved wavelength-switchable laser" in the Methods section for details). Specifically, the wavelength-switched pulse source generates Stokes pulses in four colors by every pulse pair by a four-color band-pass filter employing an array of optical intensity modulators and diffraction gratings (Supplementary Fig. 3b). The Stokes pulses are subharmonically synchronized to pump pulses, amplified by a high-power gain module (Supplementary Fig. 1c), and are combined with the pump pulses both temporally and spatially. After passing through a beam shaper that generates an elliptically shaped beam (Supplementary Fig. 1a, see "beam shaper" in the Methods section for details), the pump and Stokes pulses are focused by an objective lens (Leica, ×20, NA = 0.75) into the microchannel on the microfluidic chip (Supplementary Fig. 1d) to form a focal spot with a size of $24 \times 1\ \mu m^2$ such that the flowing cells cross the elliptical beam. The transmitted pulses are collimated by another objective lens and filtered by a short-pass optical filter. Only the pump pulses are imaged onto a Si photodetector (PD) array (Hamamatsu, S4114-35Q) (Supplementary Fig. 1e). Among the PDs in the array, 24 PDs in the middle are used to produce a series of one-dimensional (1D) SRS lines at a line rate of 48 kHz (21 μs $line^{-1}$). Each PD is equipped with electric filters and amplifiers. Its output is demodulated with a lock-in amplifier to obtain four-color SRS signals, which are digitized by an array of digitizers and FPGAs (Supplementary Fig. 1f). A 2D SRS image of an event is digitally constructed by stacking the 1D SRS lines in the flow direction in the real-time Raman image processor discussed below. This image construction is continuously repeated for multiple events. The primary difference between this ultrafast multicolor SRS microscope and our earlier SRS microscope[19] lies in their image acquisition scheme; the present SRS microscope has a parallel detection system that allows imaging at a higher flow speed than the earlier SRS microscope that has a point-scanning serial detection system.

**Real-time Raman image processor**. The real-time Raman image processor (hybrid FPGA-CPU infrastructure) is the brain of the RIACS that connects the event detector, PD array of the SRS microscope, digitizer array, multiple FPGAs, and two actuators of the dual-membrane push-pull cell sorter on the 10-Gbps all-IP network[26] to digitally process 24-parallel image detection signals and 1 event detection signal (Supplementary Fig. 4, see "architecture of the real-time Raman image processor" and "signal processing in the real-time Raman image processor" in the Methods section for details). The use of the all-IP network ensures high scalability, high throughput, high flexibility, and real-time automated operation for digital image processing and decision making. While inheriting the network architecture of the fluorescence-image-activated cell sorter[1,2], which only uses software-based image construction from the 3-channel image detectors, the real-time Raman image processor conducts the combined hardware- and software-based image construction from the array of 24-channel PDs as a consequence of the high scalability of the all-IP network. Specifically, the array of the digitizers and FPGAs performs image acquisition of one or more cells (events) based on the 24-parallel four-color SRS image signals [represented by the data acquisition (DA) node array in Fig. 1 and Supplementary Fig. 4]. Subsequently, the computer performs image construction and image analysis of the events [represented by the image analysis (IA) node in Fig. 1 and Supplementary Fig. 4], which includes generation of cell masks, linear decomposition of four-color SRS signals, classification, and sort/unsort decision making. In parallel, the other FPGAs (represented by the time management (TM) node and the sort driver in Fig. 1 and Supplementary Fig. 4) precisely determine the sort timing in the TM node based on the signal from the event detection and generates a trigger signal in the sort driver for the amplifiers followed by piezoelectric actuators for the dual-membrane pumps if a target event is identified. The 6-parallel DA nodes with the 4-channel digitizers and FPGAs realize a constant and short latency of less than 0.12 ms for the image acquisition. Consequently, all the above functions such as image construction, data transfer, image analysis, and decision making are conducted within a total of 10 ms for the real-time automated operation. The sequence of these functions is processed in parallel for consecutive events. Moreover, CPU-based software on the IA node enables the highly flexible selection of digital image analysis algorithms in the RIACS within the range of the specified image processing time. The primary difference between this real-time Raman image processor and our earlier image processor[1,2] lies in their image construction architecture; the present processor is designed such that it constructs images simply by gathering 24-parallel four-color SRS image signals sent from many data acquisition nodes whereas the earlier processor constructs images by complex signal processing such as Fourier transformation, making a significant difference in processing time (i.e., latency).

**Performance of Raman image-activated cell sorting**. We validated the basic performance of the RIACS as follows. First, we performed high-purity enrichment of polymer particles (see "preparation of polymer particles" in the Methods section for details). Figure 2a shows SRS spectra of polystyrene (PS) particles and poly(methyl methacrylate) (PMMA) particles obtained by an SRS microscope. Four wavenumbers (2899, 2954, 3006, and 3034 $cm^{-1}$) that correspond to Raman spectral peaks shown in the figure were selected and used to decompose the SRS images of the particles into two chemical species[31] (see "image analysis" and "imaging performance" in the Methods section for details) as shown in Fig. 2b. Based on the Raman spectral differences between the PS and PMMA particles, we conducted sorting of the PMMA particles from a 1/1.5 mixture of the PS and PMMA particles. As shown in Figure 2c, a scatter plot of the detected events exhibits a clear separation between two distributions attributed to the two particle species. The gating condition for sorting is also shown by a line in the figure. Figure 2d shows histograms of event rates in two separate sorting experiments with nominal throughput values of 50.2 eps and 85.6 eps (see "sorting throughput" in the Methods section for details, Supplementary Movie 1). To evaluate the sorting performance, the particles were collected at the sort and waste ports, centrifuged, and enumerated under a fluorescence microscope as shown in Fig. 2e. Figure 2f shows a theoretical estimation of the relation between throughput and sort purity at different flow speed values together with our experimental verification of it (91.6% at 50.2 eps, 81.0% at 85.6 eps) at a flow speed of 0.04 m $s^{-1}$. The theoretical estimation was obtained based on Poisson statistics of events (see "relation between throughput and purity" in the Methods section for details). Good agreement between the theoretical estimate and experimental results firmly demonstrated the real-time sorting capability with the Raman image contrast (see "evaluation of the sort purity and yield" in the Methods section for details). All RIACS experiments shown below were conducted at this flow speed to balance the sorting performance and SRS imaging sensitivity.

Next, to show the broad utility and applicability of the RIACS, we evaluated its imaging capability with diverse cell types and sizes (see "preparation of microalgal cells", "preparation of

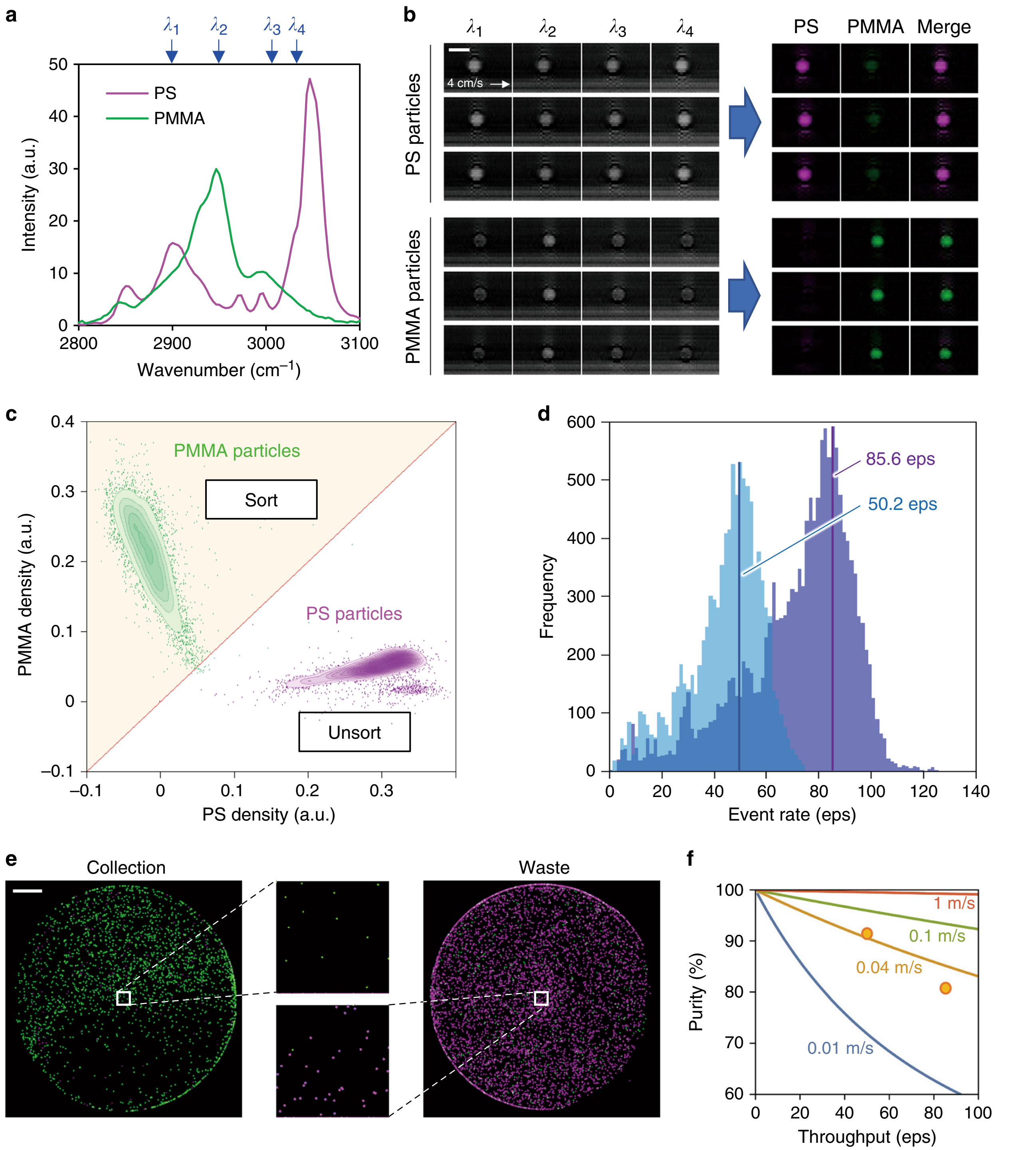


**Fig. 2 Basic performance of the RIACS. a** SRS spectra of PS and PMMA particles obtained by an SRS microscope. Four wavenumbers (2899, 2954, 3006, 3034 cm$^{-1}$) were selected and used to decompose SRS images into two chemical species. **b** SRS images of PS and PMMA particles (representative of $n=$ 9786 and $n=4063$, respectively) at the four wavenumbers and decomposed images of the particles. Scale bar, 10 μm. **c** Scatter plot of the particles in PS and PMMA intensities per particle area (referred to as PS/PMMA densities) with the sort region (yellow). **d** Histograms of event rates in two sorting experiments (nominal throughput values: 50.2 eps, 85.6 eps). **e** Fluorescence images of sorted and unsorted particles in the collection and waste tubes, respectively ($n=2$). The insets show enlarged images of the sorted and unsorted particles. Scale bar: 1 mm. **f** Theoretical estimation of the throughput-purity relation at various flow speed values together with our experimental verification (orange dots).

3T3-L1 cells", "preparation of hiPSCs", "preparation of *Chlamydomonas* sp. mutant cells", and "preparation of *Euglena gracilis* cells" in the Methods section for details). Figures 3a–d shows decomposed SRS images of various microalgal and mammalian cells obtained by the RIACS based on the SRS spectra of intracellular molecules (see Supplementary Fig. 5a–5d about our scheme for decomposing acquired SRS images of these cells into chemical images, which is essentially identical to the scheme used in Fig. 2b, and "imaging performance" in the Methods section for details), firmly demonstrating that the RIACS is capable of

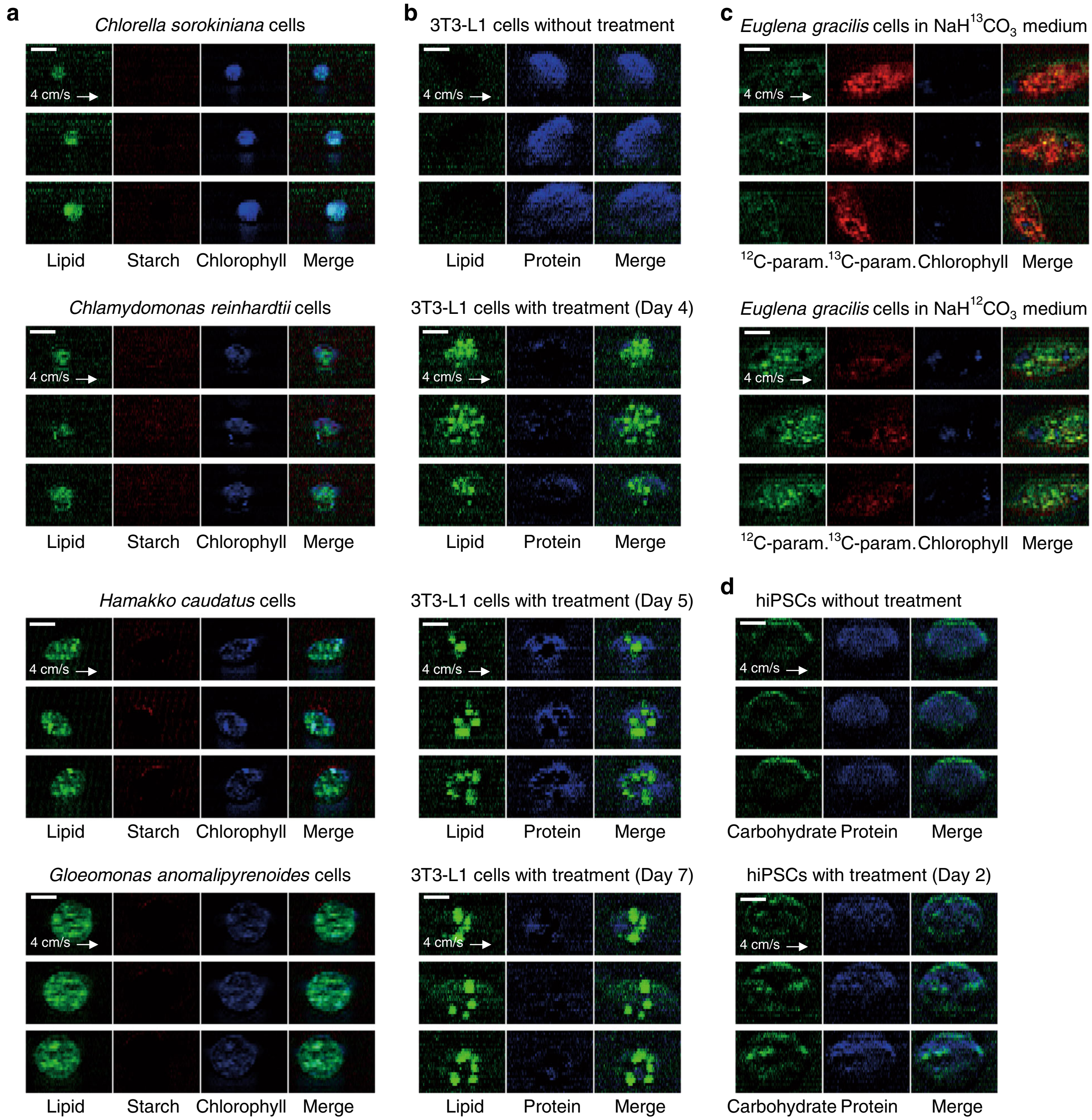


**Fig. 3 Various types of cells imaged by the RIACS.** Processing of the raw images was performed using ImageJ. Scale bars, 10 μm. **a** SRS images of various microalgal cells whose size ranges from 3 to 20 μm in cell diameter ($n = 10,348$ for *Chlorella sorokiniana* cells, $n = 12,236$ for *Chlamydomonas reinhardtii* cells, $n = 11,050$ for *Hamakko caudatus* cells, and $n = 12,793$ for *Gloeomonas anomalipyrenoides* cells). **b** SRS images of 3T3-L1 cells that gradually accumulated lipids in the cytoplasm over 7 days of treatment for inducing their differentiation into adipocyte-like cells ($n = 11,159$ for cells without treatment and $n =$ 5,892, 10,359, and 10,114 for cells with 4, 5, and 7 days of treatment, respectively). **c** SRS images of *Euglena gracilis* cells with $^{12}C/^{13}C$-isotope probing ($n =$ 5679 and 2075, respectively). **d** SRS images of hiPSCs cultivated in two different culture media for the naïve pluripotent state (with 2 days of treatment, $n = 1699$) and the primed pluripotent state (without treatment, $n = 1641$).

identifying the intracellular molecular distribution and morphological features of various types of cells. First, as shown in Fig. 3a, the RIACS was able to conduct SRS imaging of microalgal cells whose size ranged from 3 to 20 μm in cell diameter. Second, as shown in Fig. 3b, the RIACS was able to monitor the gradual accumulation of lipids within 3T3-L1 cells (an immortalized murine fibroblast-derived cell line, extensively used for studying the molecular regulation of obesity[33]) over 7 days of inducing their differentiation to adipocyte-like cells. Third, as shown in Fig. 3c, the RIACS was used in conjunction with stable isotope probing (SIP)[34,35] to differentiate two cultures of *Euglena gracilis* cells in the ability to incorporate an added stable carbon isotope probe ($^{12}C/^{13}C$) into paramylon (a carbohydrate similar to starch, produced only by the *Euglena* species[31]). Finally, as shown in Fig. 3d, the RIACS identified the differences between hiPSCs that were grown in two different culture media (one for the naïve pluripotent state and the other for the primed pluripotent state)[36] (Supplementary Fig. 6a, 6b, see "comparison between hiPSCs in two different culture media" in the Methods section for details).

**Applications of Raman image-activated cell sorting**. Among these cell types shown in Fig. 3a–d, we performed Raman

image-activated sorting of 3T3-L1 cells, *Chlamydomonas* sp. cells, and *Euglena gracilis* cells with the RIACS to validate its capability of sorting live cells in heterogeneous populations. These cell types were chosen to be able to check the sorting performance of the RIACS with fluorescence microscopy (as a ground truth provider). Each sorting experiment is described below. The experimental conditions (e.g., sorting throughput, purity, and yield) were tailored and optimized for each experiment (see "sorting experiments" in the Methods section for details).

First, we used the RIACS to conduct label-free sorting of 3T3-L1-derived adipocyte-like cells toward studying obesity, a medical condition characterized by excessive accumulation of neutral lipids in adipocytes[37,38] (Supplementary Fig. 7a, see "sorting of adipocyte-like cells" in the Methods section for details). Label-free sorting of fully differentiated adipocyte-like cells with unique spatial features (e.g., spatial distribution or localization of cytoplasmic lipid droplets, cell area, total lipid amount) is important since the differentiation and lipogenesis of adipocytes are a highly heterogeneous process[39] and overcomes difficulties of fluorescence-based technologies in analyzing cytoplasmic lipid droplets in a quantitative manner[11]. Specifically, we induced a population of 3T3-L1 cells to differentiate into adipocyte-like cells with increased heterogeneity (Fig. 4a, Supplementary Fig. 7a) and sorted adipocyte-like cells with a high-lipid density and a large spatial distribution of lipid droplets (i.e., a large standard deviation of the SRS signal intensity) within the cell (less than 1% of the total population) from the large population at a throughput of 19.1 eps (Fig. 4b). Our evaluation of the sorted and unsorted cells under a fluorescence microscope verified the sorting outcomes (Supplementary Fig. 7b). If this sorting experiment was manually performed by conventional Raman microscopy with pipetting, it would take more than 10 days[23,40], but we performed it within 10 min (about 400 times faster).

Second, we employed the RIACS to sort out rare *Chlamydomonas* sp. mutants with high-lipid productivity toward microbial breeding for cost-effective algal biofuel development[41] (see "sorting of *Chlamydomonas* sp. cells" in the Methods section for details). Here, among various types of algae for biofuel production, we used *Chlamydomonas* sp. KC4 as a sample because it is a lipid-rich strain derived from *Chlamydomonas* sp. JSC4[42], which is known to produce lipids at the highest production rate of all algae to date (~400 mg $L^{-1}$ $day^{-1}$). Sorting of lipid-rich mutants that retain their original cell size in an interference-free (i.e., staining-free) manner is important for downstream cloning and efficient microbial breeding and overcomes difficulties of fluorescence-based technologies[43] in quantitatively evaluating and isolating cells. Specifically, we applied atmospheric and room temperature plasma mutagenesis to a population of *Chlamydomonas* sp. KC4 cells (see "preparation of *Chlamydomonas* sp. mutant cells" in the Methods section for details) to induce their mutation (Fig. 4c, Supplementary Fig. 8a) and sorted out rare, highly lipid-rich mutants (about 0.3% of the total population) from the large heterogeneous population at a throughput of 36.1 eps (Fig. 4d). Our evaluation of the sorted and unsorted cells under a fluorescence microscope verified the sorting results (Supplementary Fig. 8b). If this sorting experiment was manually performed by conventional Raman microscopy with pipetting, it would take about 2 days, but we performed it within 4 min (about 700 times faster). In addition, we also performed label-free sorting of extremely rare, super-lipid-rich mutants (about 0.009% of the total population) (Supplementary Fig. 8c) and formed mutant colonies after cloning the sorted cells (Supplementary Fig. 8d), indicating a high viability of the sorted cells and their potential use for directed evolution or selective breeding.

Finally, we used the RIACS with carbon SIP to sort *Euglena gracilis* cells via $^{12}C/^{13}C$ as a tracking probe toward studying metabolism, a highly complex, dynamic, and heterogeneous process that plays an integral role in cancer biology, microbial ecology, and metabolic engineering (see "sorting of *Euglena gracilis* cells" in the Methods section for details). Since chemical properties of SIP substrates are similar to those of the original substrates, the incorporation of the stable isotope probe does not cause significant disturbance to cellular physiological functions[34,35]. In fact, SIP has been used to overcome difficulties of fluorescent probes (which are bulky and perturbative for metabolites)[9], fluorescent fusion proteins (which are not available to all types of biomolecules and cells)[10], and mass spectrometry (which only gives the average metabolic information of many cells and cannot be used to probe live cells)[44] in studying the metabolic activity of single cells in a spatiotemporally resolved and non-perturbative manner. Specifically, after assessing the RIACS' ability to separate two cultures of *Euglena gracilis* cells in $NaH^{12}CO_3$ and $NaH^{13}CO_3$, we isolated *Euglena gracilis* cells containing $^{13}C$-paramylon from a mixture of cells with different isotopologues of paramylon at a throughput of 46.0 eps (Fig. 4e, f, Supplementary Fig. 9a, 9b). Our evaluation of the sorted and unsorted cells under a fluorescence microscope (Supplementary Fig. 9c) indicates that the RIACS with SIP can be used to monitor and sort out metabolically active cells (capable of catabolizing or anabolizing molecules) from a population to study their genes and metabolic pathways. If this sorting experiment was manually performed by conventional Raman microscopy with pipetting, it would take more than 3 days, but we performed it within 6 min (about 900 times faster).

## Discussion

The capabilities of the RIACS can be enhanced in multiple directions. First, the RIACS can be used in conjunction with a fluorescence-image-activated cell sorter to increase the breadth of applications. Second, machine learning can be implemented in the real-time Raman image processor just like the intelligent image-activated cell sorter[1,2]. In this Article, we did not implement it because the cellular spatial features of interest in our experiments were not very complicated (e.g., not as complicated as the morphology of platelet aggregates which is highly diverse and may contain white blood cells as shown in our previous work[1,2]) such that classical image processing algorithms were sufficient to identify the features. In other words, machine learning algorithms are not needed unless they are required for highly accurate identification and sorting of cells since they typically consume a much longer computation time than classical image processing algorithms. Third, the use of molecular vibrations as a chemical image constrast enables super-multiplex image-activated cell sorting[45], a capability not feasible with fluorescence-based image-activated cell sorting nor non-imaging fluorescence-activated cell sorting due to the spectral overlap of fluorescence signals (typically a few to several colors at most). Since the width of vibrational signals is ~10 $cm^{-1}$, which is much narrower than the width of fluorescence signals, >20 colors can be implemented by using a palette of triple-bond-conjugated near-infrared dyes or engineered polyynes[45]. Fourth, the throughput of the RIACS can be increased by increasing the flow rate and reducing the calculation time on the real-time Raman image processor at the expense of the signal-to-noise ratio of the SRS signal acquisition, which currently limits the maximum throughput. Fifth, while in this Article we used SRS to focus on the high-frequency region (2800–3100 $cm^{-1}$), the molecular specificity of the RIACS can be significantly increased by extending it to the fingerprint region (200–1800 $cm^{-1}$) via

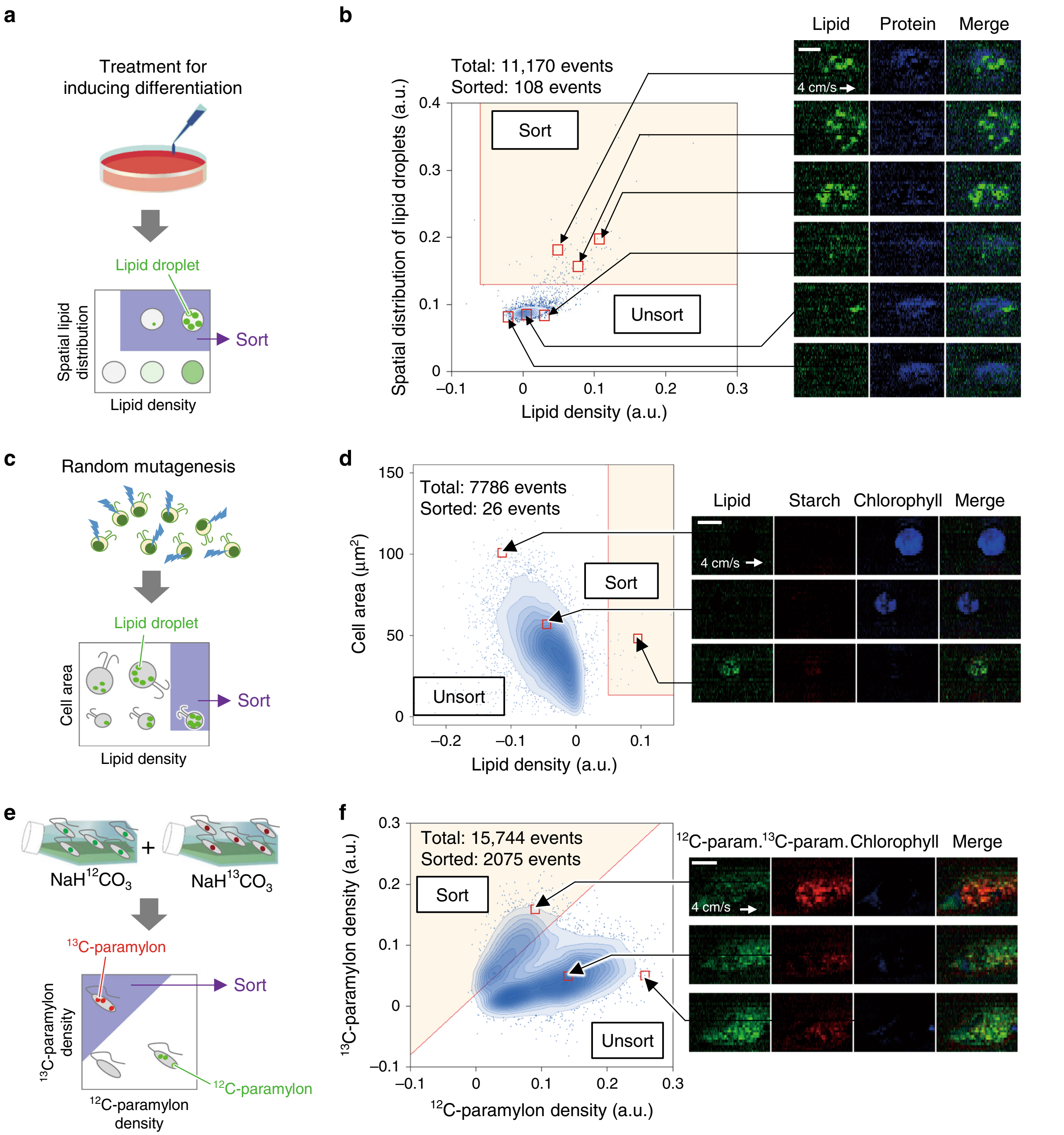


**Fig. 4 Raman image-activated sorting of various types of cells with the RIACS. a** Procedure for sorting adipocyte-like cells. **b** Scatter plot of differentiated and undifferentiated 3T3-L1 cells in the spatial distribution of intracellular lipid droplets and the lipid intensity per cell area. 3T3-L1-derived adipocyte-like cells with a large spatial distribution of cytoplasmic lipid droplets (indicated by the yellow region) were sorted by the RIACS. The insets show representative SRS images of cells in the sort ($n = 108$) and unsort regions ($n = 11{,}068$). Scale bar: 10 µm. **c** Procedure for sorting *Chlamydomonas* sp. cells. **d** Scatter plot of *Chlamydomonas* sp. mutant cells in cell area and lipid density. Highly lipid-rich, but rare (only 0.3% of the total population) *Chlamydomonas* sp. mutants (indicated by the yellow region) were sorted by the RIACS. The inset shows representative SRS images of cells in the sort ($n = 26$) and unsort regions ($n = 7760$). Scale bar: 10 µm. **e** Procedure for sorting *Euglena gracilis* cells. **f** Scatter plot of the 50/50 mixture of *Euglena gracilis* cells cultured in $NaH^{12}CO_3$ and $NaH^{13}CO_3$ in $^{12}C$- and $^{13}C$-paramylon intensity per cell area. *Euglena gracilis* cells cultured in $NaH^{13}CO_3$ (indicated by the yellow region) were sorted by the RIACS. The inset shows representative SRS images of cells in the sort ($n = 2075$) and unsort regions ($n = 13{,}669$). Scale bar: 10 µm.

high-speed coherent anti-Stokes Raman scattering spectroscopy[46], provided that its sensitivity is high enough for probing molecular vibrations in the cell. Finally, the RIACS can be directly combined with a DNA/RNA sequencing machine to enable a large statistical study of the genotype-phenotype relations of intracellular molecules[47], in particular small molecules including metabolites, which are previously difficult to label with fluorescent probes.

In this Article, the sorting throughput, number of colors, number of pixels, and cell size range were demonstrated to be ~100 eps, 4, 122 × 24 pixels, and 3–20 µm, respectively, but these

values are not fundamental physical limits and can be improved or adjusted by employing a more advanced or different architecture for the entire RIACS system as well as its major components such as the SRS microscope and real-time Raman image processor. First, the throughput can simply be boosted by increasing the flow speed of cells, but this comes at the expense of imaging sensitivity and the number of pixels in the flow direction. Therefore, an improved design for the SRS microscope with higher sensitivity and more pixels is necessary for improving the throughput although a good balance between the sensitivity, number of pixels, and throughput needs to be taken into account, depending on the application. Second, the number of colors can be increased by increasing the number of wavelength channels in the pulse-pair-resolved wavelength-switchable laser (Supplementary Figure 3b). Since the linewidth of each wavelength channel is <10 $cm^{-1}$ and the bandwidth of the SRS microscope is about 300 $cm^{-1}$, the number of colors can, in principle, be boosted to ~30, assuming that there is no spectral overlap between the consecutive channels. However, it is not meaningful to have more than several colors in this high-frequency spectral region since only several independent components can be resolved in the region even if the full spectral information is obtained. Third, the number of pixels can be increased by adding photodetection circuits, lock-in amplifiers, and digitizers (Fig. 1, Supplementary Fig. 4). Finally, while in this Article the microfluidic chip of the RIACS (i.e., microfluidic channel, cell focuser, cell sorter) were optimized for handling samples containing cells whose size range is 3–20 µm in diameter where most cell types are, a different design for the microfluidic chip is required to handle samples containing smaller cells (<3 µm), larger cells (>20 µm), or mixtures of both. For example, for applications that involve smaller cells (<3 µm) such as bacteria, other focusing techniques such as 3D hydrodynamic focusing with a narrower microchannel (instead of 2D hydrodynamic focusing used in this work), inertial focusing, deterministic lateral displacement, hydrophoretic focusing, and viscoelastic focusing[48,49] can be implemented to focus them more tightly. In summary, a careful design, adaptation, adjustment, and optimization of the system are required, depending on the application, since these parameters are not independent, but are interrelated, whereas a more sensitive SRS microscope is expected to boost the overall system performance including the throughput, number of colors, number of pixels, and cell size range.

There are numerous potential applications of the RIACS in addition to those demonstrated above. First, the RIACS is applicable to quality control of cells for cell therapies such as hematopoietic stem cell transplantation[50], mesenchymal stem cell therapy[51], CAR-T therapy[13], and hiPSC therapy[12]. The on-chip implementation of the RIACS is also effective for contamination-free quality control with high cellular viability. Second, our results shown in Fig. 3d, Supplementary Fig. 6a–6b suggest the potential to detect and isolate naïve hiPSCs from a mixture of naïve and primed hiPSCs by using cellular carbohydrate content as an indicator for efficient, low-cost regenerative medicine and more accurate disease modeling[36]. Third, the RIACS is useful for handling time-sensitive cells (e.g., platelets) for which the time-consuming process of fluorescent labeling is prohibitive. Fourth, the RIACS' ability to identify lipids in a label-free manner may be effective for isolating ovarian cancer stem cells by using desaturated lipids as a metabolic marker[52]. Fifth, our results shown in Figs. 4e–4f suggest the ability to perform pulse-chase analysis of various intracellular metabolites (e.g., protein kinase C, ubiquitin) over a prolonged period of time[53]. The RIACS with SIP can also be used to monitor and enrich metabolically active cancer cells that consume glucose at high rates and identify their genes and metabolic pathways responsible for the high activity[54]. Sixth, the RIACS may be useful for label-free liquid biopsy based on detection and sorting of CTCs in blood since they have been reported to have a much higher lipid content than blood cells[55]. This method is potentially advantageous over conventional CTC detection methods that typically require binding of fluorescent probes or metabolic fluoresent labeling, which inevitably entails fluctuations that derive from binding and labeling efficiencies, depening on the cancer stage. Finally, in addition to these potential applications, the RIACS may play an important role in cell-cycle analysis[16], synthetic-cell sorting[56], and bacterial-cell sorting[57] under a marker-free condition, provided that a different architecture for cell focusing, imaging, and sorting with application-specific performance is available. While further experimental evidence is required for validating the need for these suggested applications, expectations are high for high-content cell sorting with molecular-vibrational image contrast.

## Methods

**Preparation of polymer particles**. To evaluate the sorting performance of the RIACS, polymer particles were used. A 1/1.5 mixture of 6-µm polystyrene (PS) particles (Fluorescent Particles, Nile Red, 5.0–7.9 µm, Spherotech) and 5-µm poly (methyl methacrylate) (PMMA) particles (Fluorescent Particles, PolyAn Green, 4.7–5.3 µm, PolyAn) was injected from the sample inlet into the microfluidic chip. The concentration of the particles was tuned to be about $5 \times 10^5$ particles $mL^{-1}$ so that the event rate was set to ~100 eps in the RIACS. Note that for avoiding aggregation of the particles, 0.5%-Tween 20 was added to the sample solution.

**Preparation of microalgal cells**. Fresh water microalgae with a diverse range of cell size and morphology, *Chlorella sorokiniana* TKAC1027, *Chlamydomonas reinhardtii* TKAC1017 (NIES-2463), *Hamakko caudatus* NIES-2293, *Gloemonas anomalipyrenoides* NIES-3640, and *Euglena gracilis* NIES-48 were provided by the Microbial Culture Collection at the National Institute for Environmental Studies (NIES) (http://mcc.nies.go.jp/) and Tsuruoka Keio Algae Collection (TKAC) of T. Nakada at the Institute for Advanced Biosciences at Keio University. Microalgal cultures were grown in culture flasks (working volume: 20 mL) in a 14 h/10 h light/dark cycle with illumination at 120–140 µmol photons $m^{-2}$ $s^{-1}$ at 25 °C using AF-6 medium. To prepare carbon-accumulated cells, the culture medium was replaced with AF-6 − N medium (nitrogen nutrient omitted from AF-6).

**Preparation of 3T3-L1 cells**. 3T3-L1 cells (CL-173, ATCC) were cultured in DMEM with L-glutamine and sodium pyruvate (high glucose, 08458-45, Nacalai tesque), supplemented with 2-mercaptoethanol (198-15781, Wako), 10% fetal bovine serum bio-equivalent (EquaFETAL, EF-0500-A, Atlas), 100 units $mL^{-1}$ penicillin and 100 µg $mL^{-1}$ streptomycin (168-23191, Wako) at 37 °C, and 5% $CO_2$. The cells were cultured on culture dishes (100 mm in diameter, 150464, Thermo Fisher) and subcultured by 10-fold dilution 2–3 times per week before cultures become 60% confluent. The differentiation of 3T3-L1 cells into adipocyte-like cells was based on the method reported by Rubin et al[58]. Briefly, $3 \times 10^5$ cells in 10 mL of culture medium were seeded on the culture dishes (100 mm in diameter) 4 days prior to stimulation for the differentiation. To start the differentiation, cell culture supernatant was replaced with a fresh culture medium containing 0.5 mM 3-isobutyl-1-methylxanthine (I5879, Sigma–Aldrich), 0.25 µM dexamethasone (D4902, Sigma–Aldrich), and 10 µg $mL^{-1}$ insulin (097-06474, Wako). After 2 days, the culture supernatant was replaced with a fresh culture medium containing 1 µg $mL^{-1}$ insulin. Cells on the 5th day after starting the differentiation were used for experiments. The culture medium was replaced with appropriate fresh medium every two days.

**Preparation of hiPSCs**. IM-E1-5, one of feeder-free hiPSC lines generated from human fetal lung fibroblast IMR90[59], was used for the experiments. The cells were cultured in Essential 8 medium (E8 medium, A1517-001, Thermo Fisher) with 100 unit $mL^{-1}$ penicillin and 100 µg $mL^{-1}$ streptomycin (168-23191, Wako) on wells of 6-well plates coated with Geltrex (A14133-022, Thermo Fisher) at 37 °C and 5% $CO_2$. The hiPSC line was maintained and passaged, following the previously-reported protocol using 3 unit $ml^{-1}$ of dispase (17105-041, Thermo Fisher) in E8 medium with 10 µM Y27632 (1254, Tocris)[59]. To alter the metabolic state of the cells, a commercial medium for inducing naïve pluripotent state (ReproNaïve, RCHEMD008, ReproCell) was used according to the manufacturer's instructions. Briefly, complete ReproNaïve medium was prepared from ReproNaïve Basal Medium and ReproNaïve Supplement added with human LIF (20 ng $ml^{-1}$, 300-05, PeproTech) and Y27632 (10 µM). The cell culture medium was switched from E8 to complete ReproNaïve after passage. The complete ReproNaïve medium was used for 2 days before the experiments. In the experiments with the RIACS, hiPSCs were detached from the culture plates treated with Accutase (StemPro Accutase Cell

Dissociation Reagent, A11105-01, Thermo Fisher) at 37 °C for 5 min. The culture medium was replaced with appropriate fresh medium every day.

**Preparation of *Chlamydomonas* sp. mutant cells**. *Chlamydomonas* sp. KC4, a lipid-rich strain derived from *Chlamydomonas* sp. JSC4[42], was used as a patented strain for directed evolution or selective breeding of a super-lipid-rich mutant. A mutant library was generated by the helium-based ARTP (atmospheric and room temperature plasma) mutagenesis system ARTP-IIS (Wuxi TMAXTREE Biotechnology)[60]. The mutant library was cultured under phototrophic conditions (continuous illumination at 250 µmol photons $m^{-2}$ $s^{-1}$ with 2% $CO_2$ aeration at 30 °C) using double-deck photobioreactors (lower stage containing 2 M $NaHCO_3$/$Na_2CO_3$ to supply $CO_2$ gas and upper stage containing cell culture) and Modified Bold (MB) 6 N medium[61] for 4 days prior to the cell sorting experiments.

**Preparation of *Euglena gracilis* cells**. *Euglena gracilis* NIES-48 was cultured in culture flasks (working volume: 20 mL) in a 14 h:10 h light:dark cycle with illumination approximately 150 µmol photons $m^{-2}$ $s^{-1}$ at 25 °C using AF-6 medium. Before cultivation with $^{13}C$-stable isotope media, *Euglena gracilis* cells were grown in normal AF-6 medium for at least 3 days as preculture. The cells in the preculture were transferred to AF-6 − N medium including 20 mM of $NaH^{13}CO_3$ ($^{13}C$: 99%, Cambridge Isotope Laboratories) or $NaH^{12}CO_3$ (Wako Pure Chemical) for induction of $^{13}C$- or $^{12}C$-paramylon, respectively. After purging of air with filtered (0.22 µm) nitrogen gas and enclosing in a capped culture flask (working volume: 20 mL), cells were incubated in static conditions under continuous light illumination (~150 µmol photons $m^{-2}$ $s^{-1}$) at 28 °C for 2 days prior to cell sorting. For the evaluation of the sorting performance, the two isotope-incorporated cells were fluorescently labeled 1 h before applying them to the RIACS using different nuclear staining dyes [$^{13}C$: 20 mM of Hoechst 33342 (62249, Thermo Fisher), $^{12}C$: 5 µM of SYTO 82 (S11363, Thermo Fisher)].

**Ultrafast multicolor SRS microscope**. The main function of the ultrafast multicolor SRS microscope is to acquire four-color, blur-free SRS images of cells in a high-speed flow, which has been impossible with previous SRS microscopes[29–32,62,63]. The SRS microscope in the RIACS employs synchronized trains of pump and Stokes laser pulses whose optical frequency differences are resonant with molecular-vibrational modes (Fig. 1a-c, Supplementary Fig. 3a). The pump pulse train is generated by a picosecond mode-locked Ti:sapphire laser (Mira 900D, Coherent) with a fixed center wavelength of 790.6 nm for imaging of particles and 796.6 nm for imaging of cells, a spectral width of 0.16 nm, and a pulse repetition rate of 76 MHz. The four-color wavelength-switchable Stokes pulse train is generated by a home-built wavelength-switchable laser, which consists of a Yb fiber laser mode-locked by nonlinear polarization rotation with a center wavelength of 1030 nm, a spectral width of ~20 nm, a pulse repetition rate of 38 MHz, a four-color band-pass filter, and Yb-doped fiber amplifiers. The cavity length of the Yb fiber laser is controlled by an intracavity electro-optic modulator (NIR-MPX-LN-0.1-P-P, Photline) and a piezoelectric transducer (P-611.3 S, Physik Instrumente) on which an end mirror is fixed to achieve the precise synchronization between the pump and Stokes pulse trains[64]. The error signal of the pulse synchronization between the Ti:sapphire laser and the Yb fiber laser is obtained via two-photon absorption by a GaAsP photodiode (G1115, Hamamatsu) onto which the spatially overlapped two laser beams are focused. The error signal is used to control the cavity length (and hence the pulse repetition rate) of the Yb fiber laser via a home-made proportional-integral controller.

**Pulse-pair-resolved wavelength-switchable laser**. Fast wavelength switching of the Stokes pulses is realized by the four-color band-pass filter (Supplementary Figs. 3b, 1b). The output of the Yb fiber laser is preamplified with a Yb-doped fiber amplifier (YDFA) composed of a Yb-doped fiber (YDF) (YB501-PM, CorActive), a wavelength-division multiplexer, and a 976-nm laser diode, and then divided into four arms via three fiber couplers. In each arm, two sequential pulses out of eight pulses are picked up by an intensity modulator (IM) (NIR-MX-LN-10-P-P, Photline) and amplified by a bidirectional YDFA. The driving signal for the IM is generated by a home-made electronic circuit, which consists of a frequency divider, a binary counter, a line decoder, and output drivers (AD811, Analog Devices). The clock signal of the Yb fiber laser is input to the frequency divider so that the signal applied to each IM is temporally shifted by 53 ns. Consequently, the output wavelength is switched by every two pulses to achieve four-color SRS signal acquisition at a period of 212 ns. At the end of each arm, the pulse is collimated with a fiber collimator (OzOptics), expanded with a Galilean beam expander, and reflected back with a diffraction grating (1200 grooves per mm, 1-µm blazed, GR25-1210, Thorlabs) in a Littrow configuration to spectrally filter the pulses. The optical path length of each arm is adjusted by changing the position of the collimator. The powers of the four beams are controlled by changing the driving current of the laser diode in the bidirectional YDFA in each arm. The output of the four-color band-pass filter is obtained from one of the ports of the optical couplers and preamplified by a YDFA. The pulses are input to a high-power gain module (Supplementary Fig. 1c), which consists of a high-power photonic crystal YDFA (aeroGAIN-BASE1.3, NKT Photonics) pumped by a high-power pump source (DS3-51422-0313-K976AAHRN-27.00 W, BWT) to amplify the average power to ~3.4 W. The Stokes and pump beams are spatially and temporally overlapped via an optical delay line and a dichroic mirror.

**Beam shaper**. The beam shaper spatially shapes the pump and Stokes beams so that a linear focus orthogonal to the flow direction is formed at the optical interrogation point. As shown in Supplementary Fig. 3c, a series of achromatic spherical and cylindrical lenses transform an input circular beam into a beam horizontally collimated and vertically focused at the back aperture of the first objective lens (×20, NA = 0.75, Leica). At the focal point of the objective lens, the full width at half maximum along the long and short axes is 24 µm and 1 µm, respectively. The transmitted pump beam is detected by a vertically aligned Si photodetector (PD) array (S4114-35Q, Hamamatsu) via the second objective lens (×20, NA = 0.75, Leica), a spherical lens, the third objective lens (×20, NA = 0.45, Olympus), and imaging lenses while the transmitted Stokes beam is removed by an optical short-pass filter. In the vertical direction, the PD array surface is a conjugate plane of the optical interrogation point with a magnification factor of 1000. In the horizontal direction, the PD array surface is a Fourier plane of the optical interrogation point.

**Imaging performance**. To estimate the spatial resolution and signal-to-noise ratio (SNR) of the SRS microscope, we analyzed an SRS image of a PS particle taken by the RIACS. Supplementary Fig. 3d shows the image and its cross sections in the parallel ($x$) and perpendicular ($y$) directions to the flow, indicating that the widths of the edges are less than 2 µm in both the $x$ and $y$ directions. While this value is larger than the diffraction-limited spatial resolution of <1 µm in the $x$ direction and the pixel pitch of 1 µm in the $y$ direction, the estimation is mostly affected by the shape of the bead with a diameter of 6 µm. From the cross section in the $x$ direction, the SNR is estimated to be as high as 30. These results are consistent with the RIACS's ability to provide molecular-vibrational images of cells in a high-speed flow with a subcellular spatial resolution as shown in Fig. 3a–d. Specifically, Fig. 3a shows intracellular lipids (green) in various types of microalgal cells cultured under nitrogen deficiency conditions. In Fig. 3b, 3T3-L1 cells without treatment exhibit only the protein component (blue), while lipid droplets (green) appear after their differentiation to adipocyte-like cells. Figure 3c shows the RIACS images of *Euglena gracilis* cells cultured in $NaH^{13}CO_3$ medium and those cultured in $NaH^{12}CO_3$ medium, which exhibit granular morphology of $^{13}C$-paramylon (red) and $^{12}C$-paramylon (green), respectively. Figure 3d shows that hiPSCs exhibit a stronger carbohydrate signal after treatment, which is consistent with SRS images (Supplementary Figure 6b) taken with our SRS microscope[65].

**Design and fabrication of the microfluidic chip**. As shown in Fig. 1 and Supplementary Fig. 2a, the microfluidic chip plays the following important roles in the RIACS: (i) continuously introducing cells to the SRS microscope, (ii) delivering them from the SRS microscope to the sort point with a predictable latency, (iii) guiding the sorted/unsorted cells to the collection/waste outlets. To meet these requirements, we designed the microfluidic chip as shown in Fig. 1 and Supplementary Fig. 2a. The microfluidic chip was fabricated by using standard microelectromechanical systems (MEMS) techniques[1,2,28,66]. Since the microfluidic chip was required to be as long as the silicon wafer, we fabricated the microfluidic chip in a fabrication protocol modified from the protocol described in our previous report[1,2]. As shown in Supplementary Fig. 2a, the microfluidic chip has three layers: a base layer, a microchannel layer, and a cover layer. To form rigid microchannels that avoid potential instability due to flow-induced deformation of microchannels in the chip, we employed 250-µm thick borosilicate glass substrates for the base and cover layers and a 200-µm thick Si substrate for the microchannel layer. The use of such thin glass substrates is important to avoid physical interference with a pair of high-NA objective lenses. The microchannels including the inlets, outlets, and on-chip dual-membrane pumps were patterned by the deep reactive ion etching (DRIE) technique. We designed the microfluidic chip having a meandering microchannel for the 3D acoustic focuser[25,67], the details of which are provided in a section below. The cross-sectional dimensions of the central microchannel are 200 × 200 µm. The details of the fabrication process are described in our previous report[1,2]. First, to reduce the local stress concentration in the bonding process, we formed a grating pattern on the borosilicate glass surface of the base and cover layers by using the dry etching technique. Second, the patterned base layer and a Si substrate were bonded by anodic bonding. Third, the Si layer was etched by the DRIE technique. Fourth, the etching mask for the sandblasting process of the cover layer was patterned. Fifth, the inlets and outlets were formed by sandblasting the cover layer through the etching mask. Finally, the microfluidic chip was obtained by packaging the microchannel and the cover layer by anodic bonding.

**Optics-microfluidics integration unit**. To precisely position the central microchannel for the SRS microscope and event detector, we developed an optics-microfluidics integration unit as shown in Supplementary Fig. 2c, 2d. It mainly consists of two objective lenses and the microfluidic chip integrated with a chip holder. The objective lenses are aligned to image flowing cells at the center of the microchannel and used for monitoring the sorting process with a high-speed CMOS camera (V1211, Vision Research Inc., NJ, USA). The microfluidic chip was

sandwiched between two holding plates of the chip holder with tube connectors and piezoelectric actuators and then inserted into a slot in the integration unit in the same manner as in our previous report[1,2]. Details of the installation and optical interrogation are shown in Supplementary Fig. 2d.

**On-chip dual-membrane push-pull cell sorter**. As shown in Fig. 1 and Supplementary Fig. 2b, the microfluidic chip contains a dual-membrane sorter, which allows for precise control of the push-pull sorting actuation. The cell sorter is based on the principle of rapidly controlling the local flow at the sort point and isolating target cells from the central stream with the piezoelectrically actuated dual-membrane pumps. The configuration of the cell sorter is a modified version of the cell sorter in our previous report[28]. Each external piezoelectric actuator was set on the glass membrane fabricated as a part of the microfluidic chip. Since the glass membrane is deformed by the motion of the piezoelectric actuator, the local flow can be produced in the direction perpendicular to the cell flow. When the dual-membrane pumps are actuated in the opposite phase, high-speed local flow crossing the main microchannel is produced at the sort point as shown in Supplementary Fig. 2b. When the pump is not actuated, cells flow into the central branch of the three-branch microchannel junction, whereas when the pump is actuated by the sort trigger signal, cells flow into either the upper or lower branch (Supplementary Fig. 2b). The upper and lower channels are connected in the downstream and share the outlet. Therefore, sorted cells are collected from the outlet regardless of the direction of the local flow. This configuration is useful for high-throughput sorting because the initialization of the dual-membrane pumps is not necessary. For sorting, we applied a ramp voltage signal with an amplitude of 30 V and a rise time of 1 ms.

**Cell focusing**. As shown in Fig. 1 and Supplementary Fig. 2a, the microfluidic chip employs a 3D acoustic focuser to align cells at the center of the microchannel before arriving at the optical interrogation point (Supplementary Fig. 1a). Since the flow speed distribution in the microchannel has an approximately parabolic profile (i.e., the speed is maximum at the center), the position of cells in the cross section of the microchannel greatly affects not only the yield of image detection, but also the recovery of sorting. To obtain a reproducible and stable flow speed for cells, an ideal approach is to tightly focus the cells at the center of the microchannel. The 3D acoustic focuser meets this purpose. A 0.57-mm thick piezoelectric transducer (3.66Z20*20S-SYX, Fuji Ceramics Corporation) with a $20 \times 20$-mm area was glued on a side of the glass substrate of the microfluidic chip with an epoxy resin (7004, 3 M Japan). Acoustic focusing was accomplished by exciting both vertical and horizontal resonance modes of the $200 \times 200$ µm cross section of the microchannel by actuating the piezoelectric transducer with a sinusoidal driving signal at 3.6–3.8 MHz (depending on the weight density and temperature of the medium in the microchannel) and 75 Vpp provided by a function generator (WF1974, NF Corporation) via a high-voltage amplifier (HSA4101, NF Corporation). To improve the focusing performance, the microchannel was designed to have a meandering structure in the area under the piezoelectric transducer. The meandering structure makes the application time of the acoustic focusing longer. In addition to acoustic focusing, 2D hydrodynamic focusing can optionally be used before the acoustic focuser when needed. The hydrodynamic focuser serves to avoid cells from flowing near the microchannel walls where the effect of acoustic focusing is weak.

**Event detector**. As illustrated in Supplementary Fig. 3a, the RIACS has an event detector at the optical interrogation point to detect events (e.g., single cells, clustered cells, debris) in the view window of the SRS microscope (Supplementary Fig. 2a). A laser light used for the event detector has a spot size of approximately $6 \times 80$ µm in the flow direction and its perpendicular direction, respectively. When an object passes through the event detection spot, forward scattering (FSC) occurs due to the object and is detected by a photodetector. The FSC signal is digitized and sent to the TM node to record the arrival time at the event detection point with a unique identification (ID) number for the event. When the FSC signal intensity exceeds a predetermined threshold, the TM node generates a trigger signal for the DA node array to initiate SRS image acquisition as well as for the TM node internally to initiate the calculation of the sort time with the predetermined sort latency. The ID number is also transferred to the DA node array and then transferred to the IA node along with the SRS image of each event. After the image is analyzed and a sort/unsort decision is made, the ID number along with the sort/unsort decision is transferred back to the TM node. The TM node uses the received ID number to find the sort time of each event and then sends the sort trigger signal to the sort driver at the calculated timing.

**Architecture of the real-time Raman image processor**. To realize high scalability, high throughput, high flexibility, and real-time automated operation for digital image processing and decision making, we implemented a real-time Raman image processor based on a hybrid FPGA-CPU infrastructure on a 10-Gbps all-IP network. As shown in Fig. 1 and Supplementary Fig. 4 (a detailed version of Fig. 1), the real-time Raman image processor consists of four key components: (i) the DA node array which acquires SRS image signals, (ii) the TM node which controls the precise timing of the image acquisition and sorting, (iii) the IA node which constructs SRS images and analyzes them to make a sort/unsort decision for every event, and (iv) the all-IP network to combine all the components via the network switch to operate in real time. Details of each component are as follows. The DA node array is composed of six pairs of a 4-ch digitizer (AD9287-100EBZ, Analog Devices) and an FPGA board (SP601, Xilinx) with a 1-Gbps Ethernet port. The outputs of the 24-parallel four-color SRS signals from the lock-in amplifier array are connected to each digitizer, while the cell ID and SRS acquisition trigger signals from the TM node are connected to the FPGA boards. In the FPGA boards, digitized 4-ch SRS signals are encapsulated in UDP packets, and sent to the IA node. The timing of the data acquisition is controlled by the distributed trigger signals from the TM node. The TM node consists of an ADC board (AD9259-50EBZ, Analog Devices) and an FPGA board (KC705, Xilinx) equipped with a 1-Gbps Ethernet port. It manages the timing of the entire process including the event detection, SRS signal acquisition, image reconstruction, and sorting based on the event detection signals. The IA node receives the UDP packets from the DA nodes and makes a sort/unsort decision based on the results of the SRS image reconstruction and analysis. The IA node is equipped with a multi-core CPU, two solid state drive (SSD) storages, and 10-Gbps Ethernet ports. Consequently, the IA node stores all the information of the processed cells on the SSDs. It is important that the image analysis at the IA node is implemented as software, allowing us to employ various types of algorithms with high flexibility. All components (the DA/TM/IA nodes) are connected with the all-IP network, which is a general-purpose interconnection of a 10-Gbps Ethernet and a 1-Gbps Ethernet with an off-the-shelf network switch (XS716T-100AJS, NETGEAR). The all-IP network makes it easy to change the system configuration with high scalability simply by arranging the computing resources such as the number of the DA nodes and the number of the CPUs on the IA nodes, depending on the required processing load such as the number of the SRS signals and the event rate of the RIACS.

**Signal processing in the real-time Raman image processor**. In the real-time Raman image processor (Fig. 1, Supplementary Fig. 4), various types of signal processing including precise time management, SRS image signal acquisition, image construction, image analysis, and sort/unsort decision making are conducted by the DA nodes array, the TM node, and the IA node. Here we describe details of the signal processing flow as follows. When a cell passes through the optical interrogation point, the event detection photodiode generates an electrical pulse whose waveform is digitized and measured by the TM node. Specifically, when the intensity of the event detection signal exceeds a specified threshold value, the TM node recognizes the electrical pulse as an event. After the event is detected, a cell ID number is assigned to the event while a time stamp is attached to the ID number. After a predetermined duration (typically 5350 µs), the cell ID and trigger signals are sent to the DA node array as 2-ch electrical serial signals via a signal distributor. After the event detection signal is detected, the TM node calculates the sort time. Subsequently, all the timing information about each event is temporarily stored in a memory on the FPGA until the corresponding sort time. In parallel, the cell ID and the timing information are sent to the IA node via the UDP communication. When the digitizer in the DA nodes receives the trigger signal, the DA node begins the acquisition of the SRS signals. All pairs of the digitizers and FPGAs in the DA node array start the signal acquisition at the same time. The SRS image is built on 4-ch signals with 8-bit depth for each pair of the ADC and FPGA boards. The cell ID is received by the digital input of the FPGA boards. From the SRS image signals, each FPGA board constructs partial image strips with 4-color and 4-pixel width. The partially constructed SRS image data and the cell ID are encapsulated in UDP packets and sent to the IA node. The IA node reconstructs SRS images from the transferred data and produces 4-color SRS images with a typical field of view of $33 \times 24$ µm and an image size of $122 \times 24$ pixels. After the image construction, when the IA node receives all the UDP packets of the 4-color images with the same cell ID from the DA node array and those of the timing information from the TM node, the IA node begins image analysis. If some packets are not delivered within a specified time, the event is aborted. A sort decision is made based on the image analysis. Finally, the sort/unsort decision result with the corresponding cell ID is sent from the IA node to the TM node via the UDP communication. When the TM node receives a sort decision, a sort trigger signal is transmitted to another FPGA, which is represented by the sort driver in Supplementary Fig. 4, at the stored sort time of the cell ID. The sort driver generates a predetermined signal waveform through a DAC and consequently actuates the dual-membrane push-pull cell sorter.

**Sorting experiments**. Our protocol for cell sorting experiments is as follows. As preparatory steps, we checked and tuned the conditions of the RIACS before sorting experiments. First, to check the acoustic focusing and sorting performance, we visualized the sample stream by flowing a highly concentrated sample of 1-µm particles (~$1 \times 10^9$ particles mL$^{-1}$). While flowing the particles, we activated the sorter at 100 pulses s$^{-1}$ constantly and recorded movies with the high-speed CMOS camera. The position of the sample stream, the diameter of the sample stream, and the width of the sort window were verified to be at the center of the microchannel, 20 µm, and ~3 ms, respectively. To validate the latency settings, we monitored the trajectory of flowing 6-µm sorted particles using the high-speed CMOS camera. After these preparatory steps, we started sorting experiments. The sorting procedure of the RIACS consists of event rate tuning, preflushing, sorting, and postflushing for cell/particle collection. Before setting the prepared sample to

the sample inlet, we flushed out all particles and cells in the inlet tubing, outlet tubing, and microchannels to avoid unwanted carryovers. Then, we injected the sample into the microfluidic chip under a boost mode at a sample flow rate of ~100 μL $min^{-1}$ for 1 min, followed by a sorting mode at a sample flow rate of ~50 μL $min^{-1}$. We activated the entire RIACS to start sorting. After sorting, we injected liquid into the microchannel from the sorting ports at a flow rate of 400 μL $min^{-1}$ for 5 min and simultaneously started collecting sorted and unsorted cells. The sorting duration depends on the experiment.

**Sorting throughput**. To evaluate the throughput of the RIACS in each particle-sorting experiment, we defined event rate $e_i$ by

$$e_i = \frac{50}{t_i - t_{i-50}}, \quad (1)$$

where $t_i$ is the arrival time of the $i$-th event at the optical interrogation point and $i$ is an integer from 51 to the number of events. As shown in Fig. 2d, the event rate varied from a few eps to ~100 eps due to the operation protocol as explained in "sorting experiments" in the Methods section. Specifically, we activated the RIACS and recorded the arrival times of particles before the first particle arrived at the optical interrogation point. Therefore, the throughput of the flowing particles was initially zero, gradually increased, and reached the plateau/maximum, such that the first event rate ($e_{51}$) was equal to the minimum event rate (few eps). We used the most frequent value of the event rate as a nominal throughput.

**Relation between throughput and purity**. To derive the relation between the sort throughput and sort purity, we developed a model based on the assumption that the sort result of a particle of interest is determined by five parameters: the particle of interest itself, two neighboring particles (the particle preceding the particle of interest and the particle following the particle of interest), and two intervals (the intervals between the successive particles). Each particle is classified as a target particle or nontarget particle, while each interval is classified as an interval that is either shorter or longer than half of the sort window. The size of the sample space is $2^5 = 32$ in total. The probability of each scenario is given by the product of the probabilities of all the five parameters. The probability of each parameter is given as follows: the probability of the particle of interest being a target particle is given by the target-particle concentration $r$, while the probability of the particle of interest being a nontarget particle is given by $1 - r$; the probability of the interval $T$ of the successive particles to be shorter than half of the sort window is given by the Poisson distribution $P_p(T<\tau) = 1 - e^{-\lambda\tau}$, where $\lambda$ and $\tau$ are the throughput and half of the sort window, respectively, while the probability of the interval of the successive particles to be longer than half of the sort window is given by $1 - P_p(T<\tau)$. Here, the sort window $\tau$ is given by dividing the distance of the sorting region by the flow speed of particles.

The sample space composed of the above probabilities and the result of sorting the particle of interest in each scenario are summarized in Supplementary Table 1. The sort results are classified into four categories: true positive (TP), false positive (FP), true negative (TN), and false negative (FN). The sort results are rationalized based on the working principles of the dual-membrane push-pull cell sorter. For example, scenario no. 7 in the table is described as follows. When the particle preceding the particle of interest (Particle B in the table) arrives at the sort point, the push-pull cell sorter is activated to push the particle to the upper/lower side because it is recognized as a target particle. Simultaneously, the particle of interest is also pushed in the same direction because the interval between these two particles is shorter than half of the sort window. When the particle of interest arrives at the sort point, the push-pull cell sorter is not activated because it is recognized as a nontarget particle, such that its position is maintained in the upper/lower side. After a time delay, which is longer than half of the sort window in this scenario, the particle following the particle of interest (Particle A in the table) arrives at the sort point. At this timing, the push-pull cell sorter is activated to push the particle to the lower/upper side, which is in the opposite direction to the preceding sort action. This sort action does not affect the particle of interest because the interval between these two particles is longer than half of the sort window. Through these processes, the particle of interest, which is a nontarget particle in this scenario, flows into the upper/lower channel, resulting in a FP event.

Based on Supplementary Table 1, we calculated the sort purity using the definition

$$\mathrm{Purity}(\lambda, \tau, r) = \frac{\Sigma P_{\mathrm{TP}}}{\Sigma P_{\mathrm{TP}} + \Sigma P_{\mathrm{FP}}}, \quad (2)$$

where $\Sigma P_{\mathrm{TP}}$ and $\Sigma P_{\mathrm{FP}}$ are given by the sum of the probabilities of the scenarios that result in TP and FP events, respectively. As shown in this equation, the sort purity varies, depending on the throughput, sort window, and target-particle concentration. Moreover, since the sort window is a function of the distance of the sorting region and flow speed of particles, the sort purity varies, depending on the throughput, the distance of the sorting region, flow speed of particles, and target-particle concentration. Note that we assumed that the sort timing was predictable.

**Evaluation of the sort purity and yield**. To evaluate the sort yield and purity of sorted cells, we constructed a centrifugation-based cell counting device. A sample sorted with the RIACS was loaded into a custom glass bottom chamber with a volume of 1 mL and a viewing area of 7 mm in diameter fabricated by bonding a glass substrate (No.1 0.12–0.17 mm, 25 mm diameter, Matsunami) and a cylindrical structure made of PDMS (polydimethylsiloxane, Sylgard 184, Dow Corning), and centrifuged at 300 × $g$ for 10 min to collect cells on the glass substrate. To avoid non-specific binding of cells to PDMS, the chamber was incubated with PBS containing with 1% BSA (Bovine serum albumin solution, A9576, Sigma–Aldrich) before loading the sample. After the centrifugation, we scanned the whole viewing area with a commercially available fluorescence microscope (Ti2, Nikon instruments) equipped with a digital CMOS camera (ORCA Flash 4.0 V3, Hamamatsu Photonics) and a ×20 objective lens (S Plan Fluor ELWD ×20 DIC N1) and performed image analysis to count cells with image analysis software (NIS-Elements Ar, Nikon instech). In the PMMA particle-sorting experiment shown in Fig. 2a–g, 19 × 19 fields of vision were scanned for each of the collection and waste fluids. For verification of the performance of the sorter with particle sorting, the 5-μm PolyAn Green-labeled PMMA microparticles and 6-μm Nile Red-labeled PS particles were both detected in the GFP channel while only the PS particles were detected in the TRITC channel. Then, the particles were counted using a Spot Detection tool to measure the number and fluorescence intensities of the particles. For all particles detected in the GFP channel, each particle was identified from the ratio of the fluorescence intensities obtained in the GFP and TRITC channels. For this identification, the ratio of the intensities obtained in the GFP and TRITC channels of Nile Red-labeled PS particles was measured and used as a threshold. We denote the numbers of the sorted and unsorted particles in the collection fluid as $N_{\mathrm{PMMA,\ collection}}$ and $N_{\mathrm{PS,\ collection}}$, respectively, and those in the waste fluid as $N_{\mathrm{PMMA,\ waste}}$ and $N_{\mathrm{PS,\ waste}}$, respectively. From the enumeration results, the sort yield and purity of the sorted particles were calculated as $N_{\mathrm{PMMA,\ collection}}$ / ($N_{\mathrm{PMMA,\ collection}} + N_{\mathrm{PMMA,\ waste}}$) and $N_{\mathrm{PMMA,\ collection}}$ / ($N_{\mathrm{PMMA,\ collection}} + N_{\mathrm{PS,\ collection}}$), respectively. In all the cell sorting experiments shown in Fig. 4a–c, the evaluation method was almost the same as for the particle sorting with some specific modifications. In the case of sorting based on the Raman spectrum of lipids, the sorted cells were stained with BODIPY (0.5 mM, D3921, Thermo Fisher) for evaluation by fluorescence microscopy. The fluids recovered from the collection and waste tubes were dispensed into commercially available glass base dishes (Triple-well glass base dish, 3970-103, IWAKI) for successive centrifugal concentration (at 200 × $g$ for 3 min) since PDMS affects BODIPY staining. For experiments requiring a long time to perform (3T3-L1/BODIPY, *Euglena gracilis*/Hoechst), cells were fixed by adding 16% paraformaldehyde (15710, Electron Microscopy Sciences) at a final concentration of 4% just after the centrifugal concentration. In the 3T3-L1 sorting experiment, individual cell areas were extracted based on their bright-field images after edge-detection filtering (Gabor filter-based, NIS Elements) to estimate the number and average BODIPY signal intensity of cells in the collection and waste tubes. Judging whether individual cells have lipid droplets was made by visual inspection of the whole cell images. Cells in which lipid droplets were visually recognized were defined as sorted cells while the others as unsorted cells. In the lipid-rich *Chlamydomonas* sp. sorting experiment, individual cell areas were extracted using a bright spot detection algorithm provided by NIS Elements on the merged images of the BODIPY fluorescence and chlorophyll autofluorescence images. Among all the cells in the collection and waste tubes, 38 cells corresponding to top 0.5% in the BODIPY fluorescence intensity were defined as sorted cells while the others as unsorted cells. In the sorting experiment of stable isotope-labeled *Euglena gracilis* cells, the fluorescence signal from prestained nuclei was used as an indicator. The nucleus of *Euglena gracilis* cells stained with SYTO 82 maintained their resistance to staining against Hoechst 33342 for ~1 h while their fluorescence intensity diminished quickly after the replacement of extracellular solution from staining solution. Therefore, fluorescence images were obtained immediately after sorting for counting nuclei stained with Hoechst 33342 (that is, the number of $^{13}$C-probed *Euglena gracilis* cells), and then additional Hoechst reagent was added and incubated for more than 1.5 h to stain the nuclei of the whole cells to estimate the total number of cells by fluorescence microscopy again. For detecting nuclei, the spot detection algorithm provided by NIS Elements was used. The number of Hoechst-stained nuclei detected in the first microscopic evaluation was defined as the number of sorted cells while the difference in the number of Hoechst-stained nuclei detected between the first and second evaluation steps was defined as the number of unsorted cells.

**Comparison between hiPSCs in two different culture media**. Human stem cells are useful materials for biomedical research and cell therapies such as hematopoietic stem cell transplantation[47], mesenchymal stem cell therapy[48,68], and hiPSC therapy[12]. In the case of hiPSC therapy, label-free enrichment and applications of naïve stem cells is important for increasing the efficiency of preparing target types of differentiated cells such as retinal cells for human macular degeneraiton[12]. The RIACS has the potential to detect and isolate naïve hiPSCs from a mixture of naïve and primed hiPSCs by using cellular carbohydrate content as an indicator for efficient, low-cost regenerative medicine and more accurate disease modeling. To demonstrate this potential, we prepared two culture conditions for hiPSCs, which were grown in two different culture media as described above, and acquired their SRS images. As shown in Supplementary Fig. 6a, the difference between the two distributions in the 2D scatter plot of hiPSCs in carbohydrate and protein densities was identified, indicating the

ability of the RIACS to detect and isolate naïve hiPSCs from a mixture of naïve and primed hiPSCs. An SRS microscope was used to examine the difference in carbohydrate content between the naïve and primed hiPSCs. As shown in Supplementary Fig. 6b, the SRS microscope images revealed the significant differences in the amount and localization of carbohydrates between the naïve and primed hiPSCs.

**Sorting of adipocyte-like cells**. Obesity is a medical condition that significantly increases the risk of health problems and serious diseases such as cancer, heart disease, stroke, sleep apnea syndrome, metabolic syndrome, and diabetes and is an epidemic in the United States[37]. It is characterized by excessive accumulation of neutral lipids in adipocytes[38], which should be adequately regulated in a healthy condition. Alterations in the regulation of lipid physiology and metabolism are directly linked to development of the health problems and diseases[69]. Since the differentiation and lipogenesis of adipocytes are a highly heterogeneous process[39], enrichment of adipocytes with unique spatial features (e.g., spatial distribution or localization of cytoplasmic lipid droplets, cell area, total lipid amount) is required for detailed analysis of the cells. Here, label-free sorting with molecular-vibrational contrast is desirable since it allows for quantitative analysis of cytoplasmic lipid droplets, which is difficult with fluorescent labeling[11]. Specifically, Supplementary Fig. 7a shows a detailed schematic of our experimental procedure for Raman image-activated sorting of 3T3-L1-derived adipocyte-like cells. In step 1, a population of 3T3-L1 cells were induced to differentiate into adipocyte-like cells with increased heterogeneity. In step 2, adipocyte-like cells with a large lipid density and a large spatial distribution of lipid droplets throughout the cell from the large population were sorted out. In step 3, the sorted and unsorted cells were manually enumerated and analyzed under a fluorescence microscope with BODIPY staining. The evaluation results yielded a sort purity and yield of 74.4% and 4.8%, respectively (Supplementary Fig. 7b). The low yield is attributed to the high-purity operation mode. A further molecular analysis of the sorted cells may lead to identification of genes and regulatory pathways that are responsible for their heterogeneous lipogenesis.

**Sorting of *Chlamydomonas* sp. cells**. The development of reliable, sustainable, and economical sources of alternative fuels is an important, but challenging goal for the world. As an alternative to liquid fossil fuels, algal biofuel is expected to be one of the most promising solutions for alleviating the detrimental effects of global warming since algae absorb atmospheric $CO_2$ via photosynthesis[31,41,70]. This is the only process that can convert $CO_2$ into organic compounds with high energy content and can, hence, provide a source for sustainable liquid transportation fuel. Furthermore, algae require minimal environmental resources because they can be grown even in saline and wastewater[71]. However, algal biofuel must overcome several critical technical challenges, particularly in selective breeding, before it is deployed in the competitive fuel market. For this reason, it is critical to rapidly identify and isolate highly lipid-rich algal cells that can produce and accumulate lipids without increasing their cell size in an interference-free (e.g., staining-free) manner for downstream cloning and efficient microbial breeding. To date, massive screening of high-lipid *Chlamydomonas* mutants has been performed by fluorescence-activated cell sorting[43], but not in a label-free manner. Specifically, Supplementary Fig. 8a shows a detailed schematic of our experimental procedure for Raman image-activated sorting of *Chlamydomonas* sp. cells. In step 1, atmospheric and room temperature plasma mutagenesis was applied to a population of *Chlamydomonas* sp. KC4 cells to induce their mutation. In step 2, rare, highly lipid-rich mutants (about 0.3% of the total population or 26 events) were sorted out of the large heterogeneous population (7,786 events) in the specified gating condition. In step 3, the sorted and unsorted cells were collected with the centrifugation-based cell counting devices, followed by manual enumeration and analysis under a fluorescence microscope with BODIPY staining. The evaluation results yielded a sort purity and yield of 32.1% and 46.2%, respectively (Supplementary Fig. 8b). Furthermore, to find extremely rare, super-lipid-rich mutants, we sorted extremely lipid-rich mutants (about 0.009% of the total population or only 20 events) from a very large heterogeneous population (220,152 events) (Supplementary Fig. 8c). We directly dropped the sorted cells onto an agar plate, on which we identified 25 colonies after cultivation (Supplementary Fig. 8d), indicating the RIACS' ability to sort a large number of cells without causing fatal damage to the cells. For application to practical biofuel production, it is necessary to further investigate the temporal stability of lipid-rich mutants in terms of intracellular lipid concentration.

**Sorting of *Euglena gracilis* cells**. Metabolism is a highly complex, dynamic, and heterogeneous process and plays an integral role in a broad range of fields such as cancer biology, microbial ecology, and metabolic engineering[72]. Disturbances in cell metabolism often lead to various prevalent diseases such as cancer, inflammation, and cardiovascular disease, whereas microbial metabolism is of prominent significance in virtually all ecosystems including the natural environment and human body. A better understanding of the dynamics and heterogeneity of metabolism requires an analytical tool to study the metabolic activity of single cells in a spatiotemporally resolved and non-perturbative manner, but conventional technologies are insufficient for meeting this requirement. SIP[34,35] on the RIACS platform offers a unique approach to studying metabolic activity at the single-cell level. By tracking Raman shifts accompanied by cellular incorporation of an added stable isotope probe enables evaluation and comparison of intracellular metabolic activity at a given time. In fact, stable isotopes have been used as non-invasive tracers for monitoring the intracellular dynamic process of glucose, amino acids, and lipid precursors[46,73]. Specifically, Supplementary Fig. 9a shows a detailed schematic of our experimental procedure for Raman image-activated sorting of *Euglena gracilis* cells. In step 1, *Euglena gracilis* cells were cultivated in two different culture media composed of $NaH^{12}CO_3$ and $NaH^{13}CO_3$. In step 2, their incorporation of the stable carbon isotope into paramylon (a carbohydrate similar to starch, produced only by the *Euglena* species) was monitored. As shown in Supplementary Fig. 9b, the separation between the two distributions indicates the RIACS' ability to probe and isolate cells containing $^{13}C$-paramylon from a mixture of cells with different isotopologues of paramylon. In step 3, the two populations were mixed at the 50/50 mixing ratio. In step 4, cells containing $^{13}C$-paramylon were sorted out of the mixed population in the specified gating condition. Our enumeration and analysis of the sorted and unsorted cells under a fluorescence microscope with SYTO 82 and Hoechst staining for identifying $^{12}C$-paramylon and $^{13}C$-paramylon, respectively, gave a sort purity and yield of 85.4% and 36.2%, respectively (Supplementary Fig. 9c).

**Image analysis**. For each event, a binary image mask was created from its selected color image, followed by the extraction of various image features from the four-color SRS images[74]. The image analysis algorithm was coded with OpenCV on C++. First, the pixel size was normalized to a square, leading to an image size of $122 \times 89$ pixels. Specifically, after applying a background correction, a Gaussian filter, and a top-hat filter to the images to remove noise, Canny edge detection was performed on the selected color image for detecting the edge of the event (e.g., particle, cell) from the background, and then contours were obtained as a binary image. Based on the extracted contours, the binary image mask of the cell was obtained after morphological analysis operations such as erosion, dilation, flood-fill, and convex-hall. The linearly decomposing algorithm[31] was used in parallel to obtain the decomposed images from four-color SRS images. Then, the mask was used to extract morphological features of the event including area and shape as well as intensity information including the average, maximum, and standard deviation of the decomposed images. It is important to note that the RIACS can accept any image feature such as perimeter, circularity, and chemical localization as long as the total signal processing time fits within the sort latency. Separately, offline processing for quantification and statistical analysis was implemented with Pillow on Python and R, respectively. ImageJ (NIH) was used to produce colored images and merged images shown in Fig. 2b, Fig. 3, Fig. 4b, Fig. 4d, and Fig. 4 f from the decomposed images.

The details of the linearly decomposing algorithm are explained as follows. We denote the measured four-color SRS data at the $j$th pixel as $\mathbf{d}_j$ and the spectral bases of the $i$th constituent as $\mathbf{s}_i$ ($i = 1, 2, \ldots n$), where $n$ is the number of constituents, and assume that $\mathbf{d}_j$ is decomposed into the linear summation of $\mathbf{s}_i$, that is, $\mathbf{d}_j = \sum_i c_{ji}\mathbf{s}_i$, where the coefficients $c_{ji}$ are the concentrations of the corresponding constituents. By multiplying the pseudo matrix inversion of $[\mathbf{s}_1\mathbf{s}_2 \cdots \mathbf{s}_n]$ to $\mathbf{d}_j$, we obtain $c_{ji}$. Supplementary Fig. 5a–5d show four-color SRS spectral bases used in the imaging and sorting experiments.

**Reporting summary**. Further information on research design is available in the Nature Research Reporting Summary linked to this Article.

## Data availability

The source data underlying Figs. 2c, 2d, 4b, 4d, and 4f are available in the custom codes[74]. An additional dataset that supports findings in this study is available upon reasonable request to the corresponding author.

## Code availability

The custom codes for image analysis and sort/unsort decision making, and for data analysis for the figures are available from a public repository[74].

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

## Acknowledgements

This work was primarily supported by the ImPACT program of the Council for Science, Technology, and Innovation (Cabinet Office, Government of Japan) and partly supported by KISTEC, JSPS Core-to-Core Program, White Rock Foundation, and Precise Measurement Technology Promotion Foundation. N.N. is an ISAC Marylou Ingram Scholar.

## Author contributions

K.G. conceived the concept of the RIACS. T.Ii., Y.Ki., Y.Su., D.D., K.Hiram., Y.I., C.L., H.M., S.T., and Y.Oz. designed and constructed the optics including the SRS microscope and event detector. A.I., S.S., F.A., T.Hay., and Y.Kas. designed, fabricated, and evaluated the microfluidic chip, fluidic system, hydrodynamic and acoustic focusers, dual-membrane push-pull sorter, and optics-microfluidics integration unit. H.T., M.O., K.Hirak., M.I., T.O., T.Se., and T.Su. designed and implemented the hardware and software of the event detector, real-time Raman image processor, and sort driver. N.N., T.Ii., A.I., Y.Ki., S.S., Y.Su., H.T., M.O., K.Hirak., Y.Kas., T.Su., and Y.Oz. integrated the parts of the RIACS. T.Ii., A.I., M.Yamag., Y.Ki., Y.I., H.Ka., Y.Sh., N.S., S.T., and T.Su. conducted the sorting experiments and data analysis. N.N., T.Ii., A.I., M.Yamag., Y.Ki., H.F., T.Has., Y.Hosh., T.It., M.K., Y.Kat., S.M., A.N., K.N., K.Sh., H.W., T.Y., M.Yaz., Y.Yo., S.Uem., and Y.Oz. designed the experiments. H.Ko. and T.Su. designed and implemented the data analysis software. T.A., M.H., T.It., and Y.Kat. prepared the microalgal cells and analyzed the results. M.Yamag., T.A., T.It., and S.Uen. prepared the cell line samples and analyzed the results. N.N., Y.Hosh., T.It., A.N., Y.Hoso., S.Uem., T.Su., Y.Oz., and K.G. supervised the work. N.N., A.I., T.It., D.D.C., S.Uem., T.Su., Y.Oz., and K.G. wrote the manuscript. T.Ii., M.Yamag., and Y.Ki. provided assistance with the figures and supplemental materials.

## Competing interests

Y.Su., Y.Oz., and K.G. are inventors of a patent application covering SRS imaging with a wavelength-switched laser. S.S., F.A., and T.Hay. are inventors of a patent application covering the dual-membrane push-pull cell sorter. N.N., T.Su., and K.G. are shareholders of CYBO. The remaining authors declare no competing interests.

## Additional information








Nao Nitta[1,2,3], Takanori Iino[4,28], Akihiro Isozaki[1,5,28], Mai Yamagishi[6,28], Yasutaka Kitahama[1,28], Shinya Sakuma[7,28], Yuta Suzuki[4], Hiroshi Tezuka[8], Minoru Oikawa[9], Fumihito Arai[7], Takuya Asai[4], Dinghuan Deng[4], Hideya Fukuzawa[10], Misa Hase[1], Tomohisa Hasunuma[11,12], Takeshi Hayakawa[13], Kei Hiraki[1], Kotaro Hiramatsu[1], Yu Hoshino[14], Mary Inaba[8], Yuki Inoue[4], Takuro Ito[1,2], Masataka Kajikawa[10], Hiroshi Karakawa[1], Yusuke Kasai[7], Yuichi Kato[12], Hirofumi Kobayashi[1], Cheng Lei[1,15], Satoshi Matsusaka[16,17], Hideharu Mikami[1], Atsuhiro Nakagawa[18], Keiji Numata[19], Tadataka Ota[1], Takeichiro Sekiya[4], Kiyotaka Shiba[20], Yoshitaka Shirasaki[6], Nobutake Suzuki[6], Shunji Tanaka[4], Shunnosuke Ueno[1], Hiroshi Watarai[21], Takashi Yamano[10], Masayuki Yazawa[22], Yusuke Yonamine[23], Dino Di Carlo[1,24,25,26], Yoichiroh Hosokawa[27], Sotaro Uemura[6], Takeaki Sugimura[1,2,3], Yasuyuki Ozeki[4] & Keisuke Goda[1,2,15,24]✉

[1]Department of Chemistry, The University of Tokyo, Tokyo 113-0033, Japan. [2]Japan Science and Technology Agency, Kawaguchi 332-0012, Japan. [3]CYBO, Tokyo 101-0022, Japan. [4]Department of Electrical Engineering and Information Systems, The University of Tokyo, Tokyo 113-8656, Japan. [5]Kanagawa Institute of Industrial Science and Technology, 705-1 Shimoimaizumi, Ebina, Kanagawa 243-0435, Japan. [6]Department of Biological Sciences, The University of Tokyo, Tokyo 113-0033, Japan. [7]Department of Micro-Nano Mechanical Science and Engineering, Nagoya University, Nagoya 464-8601, Japan. [8]Department of Creative Informatics, The University of Tokyo, Tokyo 113-0033, Japan. [9]Natural Sciences Cluster, Sciences Unit, Kochi University, Kochi 780-8520, Japan. [10]Graduate School of Biostudies, Kyoto University, Kyoto 606-8502, Japan. [11]Graduate School of Science, Technology and Innovation, Kobe University, Kobe 657-8501, Japan. [12]Engineering Biology Research Center, Kobe University, Kobe 657-8501, Japan. [13]Department of Precision Mechanics, Chuo University, Tokyo 192-0393, Japan. [14]Department of Chemical Engineering, Kyushu University, Fukuoka 819-0395, Japan. [15]Institute of Technological Sciences, Wuhan University, 430072 Wuhan, China. [16]Department of Clinical Research and Regional Innovation, University of Tsukuba, Ibaraki 305-8577, Japan. [17]Department of Gastroenterology, Cancer Institute Hospital, Japanese Foundation for Cancer Research, Tokyo 135-8550, Japan. [18]Department of Neurosurgery, Graduate School of Medicine, Tohoku University, Sendai 980-8577, Japan. [19]Biomacromolecules Research Team, RIKEN Center for Sustainable Resource Science, Wako 351-0198, Japan. [20]Division of Protein Engineering, Cancer Institute of Japanese Foundation for Cancer Research, Tokyo 135-8550, Japan. [21]Institute of

Medical Science, The University of Tokyo, Tokyo 108-8639, Japan. [22]Department of Rehabilitation and Regenerative Medicine and Pharmacology, Columbia University, New York 10032, USA. [23]Research Institute for Electronic Science, Hokkaido University, Sapporo 001-0021, Japan. [24]Department of Bioengineering, University of California, Los Angeles, CA 90095, USA. [25]Department of Mechanical Engineering, University of California, Los Angeles, CA 90095, USA. [26]California NanoSystems Institute, University of California, Los Angeles, CA 90095, USA. [27]Division of Materials Science, Nara Institute of Science and Technology, Ikoma 630-0192, Japan. [28]These authors contributed equally: Takanori Iino, Akihiro Isozaki, Mai Yamagishi, Yasutaka Kitahama, Shinya Sakuma. ✉email: goda@chem.s.u-tokyo.ac.jp