## [Peer Review File · Nature Communications]

Reviewers' Comments:

Reviewer #1:

Remarks to the Author:

This manuscript describes the first Raman imaging-activated cell sorting platform based on multicolour SRS imaging as well as applications of which to various cell types. The manuscript is well written and the authors should address a few comments as follows:

1. In the introduction, the authors claimed “~600 times richer information content by providing Raman images than (non-imaging) activated cell sorting”. Although lots of spontaneous Raman-activated cell sorting are based on resonance Raman, one should not neglect that one spontaneous Raman spectrum (as well as stable-isotope labelled Raman spectrum) contains hundreds of intensity information and far richer than a coherent Raman spectrum.
2. The numbers listed in the introduction should be carefully justified.
3. The first part of the RESULTS section is lengthy and with lots of technical details. The authors should consider putting some of it into METHODS.
4. In Figure 3 and 4, the chosen wavenumbers of different channels should be labelled on the graph.

In the ABSTRACT and INTRODUCTION, the authors specified the strength of image-activated cell sorting in the connection between cell-level analysis and gene-level analysis. However, this strength was not illustrated in the experiments. The authors should comment it in the DISCUSSION.

Reviewer #2:

Remarks to the Author:

The manuscript by Nitta et al. present a Raman image-activated method for real-time cell sorting. This work is the result of a tremendous amount of effort from a large and diverse team. The results are potentially impactful, especially given the label-free chemical specificity that the method delivers. I have several comments, mostly minor, as follows.

1. It would be beneficial to detail better the innovation with respect to the previous work published in PNAS 2019.
2. Can the authors comment on the fundamental limits in terms of throughput, spectral multiplexing, etc? What allowed the authors to produce this boost in speed and what keeps them from going further?
3. The main text and supplemental info suggest a rather complex system. Maybe some readers might be interested to see an estimate the cost for this instrument compared to a regular flow cytometer.
4. On page 4, the authors use some multipliers to describe performance of their method: “100 times higher throughput”, “600 times richer information”, “60,000 times more powerful”. To make these numbers meaningful, the authors should define quantitatively “throughput”, “powerful”, and “rich”.
5. “Eps” should be defined before its first appearance.
6. There are some sentences that could use rewording, e.g., “were straightforward and not very complicated”, “wavelength-switched pulse source generates Stokes pulses whose wavelength is switched”, etc
7. This statement in the abstract seems a bit too strong “imaging-based cell picking has revolutionized single-cell biology”. Would the authors say that image-based flow cytometry is now the norm? Or is it extremely promising for now?

In sum, this is an excellent piece of work, with complex imaging, computational, and analysis components, demonstrated carefully on various cell types.

Reviewer #3:

Remarks to the Author:

Key results:

In this manuscript, the authors demonstrated "Raman" image activated cell sorting by integration of ultrafast multicolour stimulated Raman Scattering microscopy, digital image processing, and a microfluidic cell sorter. The system has been applied to sort various types of cells with different treatments, illustrating a maximum throughput of ~100 events per second.

Originality and significance:

The concept of this work is not new. It is the combination of the previous work from the same group, i.e. image-activated cell sorting (Nitta, Cell 2018) and label-free chemical imaging flow cytometry by high-speed multicolour stimulated Raman Scattering (Suzuki, PANS 2019). Most of the technical platforms (including microfluidic sorter) in this work have been illustrated in their previous publications.

As the authors claimed, there is some technical development in terms of imaging acquisition scheme (Line 169) and image construction architecture (line 198). The current implementation uses a parallel detection system and constructs images from 24-parallel 4-colour SRS signals. These perhaps have shortened on-line processing time, which is necessary for high throughput sorting. Clearly, the group are experts in stimulated Raman Scattering (SRS) and imaging-activated cell sorting. This work shows the capability of SRS imaging for high throughput cell sorting.

However, the title of "Raman image activated cell sorting" is misleading. Generally, Raman often refers to spontaneous Raman spectroscopy. However, the work is based on Stimulated Raman Scattering. This is not mentioned anywhere in the title and abstract.

The authors seem to overstate the advantage of their approach by saying "our Raman image-activated cell sorter (RIACS) offers not only ~100 times higher throughput, but also ~600 times richer information content by providing Raman images ..., making the RIACS ~60,000 times more powerful in terms of both population-level and single-cell-level information contents than the state-of-the-art technology". This "60,000 times more powerful.." doesn't make any sense. Every approach has pros and cons. Spontaneous Raman spectroscopy is slow, however, it offers fine spectral resolution and is well-established in biological communities. RIACS only uses 4-colour SRS imaging, which is essential for its fast spectra acquisition that comes at the expense of spectral resolution. In addition, Figures 3, 4 and supplementary figure 9 show the spectral features are sufficient to differentiate cells, and the spatial information contained in the images does not seem to add significantly to this.

Methodology, statistics and others:

Although the manuscript is well written, there are still some issues as detailed below:

- 1) The authors didn't clarify where the claimed ~100 events per second comes from. The calculation of the evident rates e_i in the "sorting throughput" section is confusing. Do the histograms of event rates in Figure 2d suggest the event rates vary in a single experiment?
- 2) In the "Sorting experiment" section, the authors mention that the diameter of 1 μ m particles sample stream is 20 μ m. This indicates poor focusing for small particles. The authors should discuss the limitations of the system for sorting small objectives, since the potential applications proposed in the discussion include small sized cells (e.g. synthetic-cell sorting and bacterial-cell sorting).
- 3) If the authors sort particles one by one, they should provide evidence.
- 4) For the cell sorting experiments, no information is given about statistics. It is not clear whether the purity and yield data are average values or from a single experiment.
- 5) In figure 2b, the scale bar is missing.

We are grateful to the Reviewers for taking the time to review our manuscript and give us their valuable comments. We have taken all the comments into consideration and have made appropriate changes to the manuscript. Our point-by-point response appears below, in which we first echo each Reviewer's comments (shown in italic) and then respond to them. Our revisions are shown in the revised manuscript in **red**.

TO REVIEWER #1

Reviewer #1's comment #1:

This manuscript describes the first Raman imaging-activated cell sorting platform based on multicolour SRS imaging as wells as applications of which to various cell types. The manuscript is well written and the authors should address a few comments as follows:

Authors' response:

We thank the Reviewer for recognizing our work and giving us the positive comment.

Reviewer #1's comment #2:

In the introduction, the authors claimed “~600 times richer information content by providing Raman images than (non-imaging) activated cell sorting”. Although lots of spontaneous Raman-activated cell sorting are based on resonance Raman, one should not neglect that one spontaneous Raman spectrum (as well as stable-isotope labelled Raman spectrum) contains hundreds of intensity information and far richer than a coherent Raman spectrum.

Authors' response:

We thank the Reviewer for the comment. We agree with him/her that perhaps the original statement was somewhat misleading as it could be interpreted unfair to researchers who work on spontaneous Raman spectroscopy. To address his/her comment, we have revised the statement as follows (page 4, paragraph 1): “As a result, our Raman image-activated cell sorter (RIACS) offers ~100 times higher sorting throughput as well as spatial resolution by providing Raman images in comparison to (non-imaging) Raman-activated cell sorting that provides only one-dimensional (1D) Raman signal intensities with a moderate throughput of ~1 eps.”

Reviewer #1's comment #3:

The numbers listed in the introduction should be carefully justified.

Authors' response:

We thank the Reviewer for the comment. To address it, we have clarified the statement that contains the numbers in the revised manuscript as follows (page 4, paragraph 1; identical to our response to comment #2): “As a result, our Raman image-activated cell sorter (RIACS) offers ~100 times higher sorting throughput as well as spatial resolution by providing Raman images in comparison to (non-imaging) Raman-activated cell sorting that provides only one-dimensional (1D) Raman signal intensities with a moderate throughput of ~1 eps.”

Reviewer #1's comment #4:

The first part of the RESULTS section is lengthy and with lots of technical details. The authors should consider putting some of it into METHODS.

Authors' response:

We thank the Reviewer for the suggestion. We have separated the first subsection of the Results section into three subsections, such that the first subsection discusses the big picture of the RIACS while the last two describe the functionality of the primary components, namely the ultrafast multicolor stimulated Raman scattering microscope and real-time Raman image processor, which can be skipped by readers who may not be interested in the technical details of the microscope and image processor. We hope this revised structure will make it easier to read.

Reviewer #1's comment #5:

In Figure 3 and 4, the chosen wavenumbers of different channels should be labelled on the graph.

Authors' response:

We thank the Reviewer for the comment. Actually, the colors in the images shown in Figures 3 and 4 do not represent the wavenumbers of the SRS channels, but indicate the species of intracellular molecules obtained by converting the SRS-wavenumber-resolved images into the molecular-species-resolved images using the spectral decomposition scheme shown in Supplementary Figure 5. This process is identical to that shown in Figure 2b. In the revised manuscript, we have added the following sentence to clarify this point (page 7, paragraph 2): “see Supplementary Figures 5a-5d about our scheme for decomposing acquired SRS images of these cells into chemical images, which is essentially identical to the scheme used in Figure 2b”.

Reviewer #1's comment #6:

In the ABSTRACT and INTRODUCTION, the authors specified the strength of image-activated cell sorting in the connection between cell-level analysis and gene-level analysis. However, this strength was not illustrated in the experiments. The authors should comment it in the DISCUSSION.

Authors' response:

We thank the Reviewer for the suggestion. To address it, we have added the following text to the Discussion section (page 10, paragraph 1): “Finally, the RIACS can be directly combined with a DNA/RNA sequencing machine to enable a large statistical study of the genotype-phenotype relations of intracellular molecules⁴⁷, in particular small molecules including metabolites, which are previously difficult to label with fluorescent probes.”

TO REVIEWER #2**Reviewer #2's comment #1:**

The manuscript by Nitta et al. present a Raman image-activated method for real-time cell sorting. This is work is the result of a tremendous amount of effort from a large and diverse team. The results are potentially impactful, especially given the label-free chemical specificity that the method delivers. I have several comments, mostly minor, as follows.

Authors' response:

We thank the Reviewer for recognizing our work and giving us the positive comment.

Reviewer #2's comment #2:

It would be beneficial to detail better the innovation with respect to the previous work published in PNAS 2019.

Authors' response:

We thank the Reviewer for the suggestion. The major innovation in this work compared with the PNAS 2019 work is the ability to conduct real-time sorting of cells based on their SRS images. More specifically, as we stated on page 4 in the original manuscript, “Previous work has shown the ability to sort cells based on

fluorescence images [1-3] or acquire Raman images in continuous flow [19; PNAS paper], but no previous work has shown the ability to acquire and process Raman images rapidly enough to sort cells. This requires significant innovations in Raman image acquisition, digital image processing, and seamless integration of them with fluidic and mechanical devices into a complete system, which we demonstrate here to achieve not only Raman image-activated cell sorting for the first time, but also at unprecedented rates of ~100 events per second (eps).” Although we understand that the innovations may not be very clear to some readers because they are highly interdisciplinary, we feel that the details of the Raman image acquisition, digital image processing, and seamless integration of them with fluidic and mechanical devices into the complete system are sufficiently provided in the Results and Methods sections.

Reviewer #2’s comment #3:

Can the authors comment on the fundamental limits in terms of throughput, spectral multiplexing, etc? What allowed the authors to produce this boost in speed and what keeps them from going further?

Authors’ response:

We thank the Reviewer for the suggestion. Our brief answer to the question of how to produce the boost in speed is, as we state in the original manuscript, that “This requires significant innovations in Raman image acquisition, digital image processing, and seamless integration of them with fluidic and mechanical devices into a complete system, which we demonstrate here to achieve not only Raman image-activated cell sorting for the first time, but also at unprecedented rates of ~100 events per second (eps).” As for the second part of the Reviewer’s question, we have added the following paragraph and related references to the Discussion section in the revised manuscript to discuss the fundamental limits on the sorting throughput, number of colors, etc. (page 10, paragraph 2): “In this Article, the sorting throughput, number of colors, number of pixels, and cell size range were demonstrated to be ~100 eps, 4, 122×24 pixels, and 3-20 μm , respectively, but these values are not fundamental physical limits and can be improved or adjusted by employing a more advanced or different architecture for the entire RIACS system as well as its major components such as the SRS microscope and real-time Raman image processor. First, the throughput can simply be boosted by increasing the flow speed of cells, but this comes at the expense of imaging sensitivity and the number of pixels in the flow direction. Therefore, an improved design for the SRS microscope with higher sensitivity and more pixels is necessary for improving the throughput although a good balance between the sensitivity, number of pixels, and throughput needs to be taken into account, depending on the application. Second, the number of colors can be increased by increasing the number of wavelength channels in the pulse-pair-resolved wavelength-switchable laser (Supplementary Figure 3b). Since the linewidth of each wavelength channel is $<10 \text{ cm}^{-1}$ and the bandwidth of the SRS microscope is about 300 cm^{-1} , the number of colors can, in principle, be boosted to ~30, assuming that there is no spectral overlap between the consecutive channels. However, it is not meaningful to

have more than several colors in this high-frequency spectral region since only several independent components can be resolved in the region even if the full spectral information is obtained. Third, the number of pixels can be increased by adding photodetection circuits, lock-in amplifiers, and digitizers (Figure 1, Supplementary Figure 4). Finally, while in this Article the microfluidic chip of the RIACS (i.e., microfluidic channel, cell focuser, cell sorter) were optimized for handling samples containing cells whose size range is 3-20 μm in diameter where most cell types are, a different design for the microfluidic chip is required to handle samples containing smaller cells ($<3 \mu\text{m}$), larger cells ($>20 \mu\text{m}$), or mixtures of both. For example, for applications that involve smaller cells ($<3 \mu\text{m}$) such as bacteria, other focusing techniques such as 3D hydrodynamic focusing with a narrower microchannel (instead of 2D hydrodynamic focusing used in this work), inertial focusing, deterministic lateral displacement, hydrophoretic focusing, and viscoelastic focusing can be implemented to focus them more tightly. In summary, a careful design, adaptation, adjustment, and optimization of the system are required, depending on the application, since these parameters are not independent, but are interrelated, whereas a more sensitive SRS microscope is expected to boost the overall system performance including the throughput, number of colors, number of pixels, and cell size range.”

Reviewer #2's comment #4:

The main text and supplemental info suggest a rather complex system. Maybe some readers might be interested to see an estimate the cost for this instrument compared to a regular flow cytometer.

Authors' response:

We thank the Reviewer for the suggestion. We agree that some readers might show interest in seeing the cost of the RIACS, but we feel reluctant to provide it for the following reasons: (1) it is essentially difficult to estimate the total cost of the entire RIACS system without including the labor cost because it contains numerous custom-made components; (2) the RIACS system was optimized in terms of performance, but not in cost, and therefore, it is not fair to compare in commercial value (cost) between the prototype and an already established (commercial) flow cytometer; (3) this paper is for a pure scientific purpose, not directly for a commercial purpose.

Reviewer #2's comment #5:

On page 4, the authors use some multipliers to describe performance of their method: “100 times higher throughput”, “600 times richer information”, “60,000 times more powerful”. To make these numbers meaningful, the authors should define quantitatively “throughput”, “powerful”, and “rich”.

Authors' response:

We thank the Reviewer for the comment. To address it, we have defined the throughput and clarified the statement in the revised manuscript as follows (page 4, paragraph 1): “As a result, our Raman image-activated cell sorter (RIACS) offers ~100 times higher sorting throughput as well as spatial resolution by providing Raman images in comparison to (non-imaging) Raman-activated cell sorting that provides only one-dimensional (1D) Raman signal intensities with a moderate throughput of ~1 eps.”

Reviewer #2's comment #6:

“Eps” should be defined before its first appearance.

Authors' response:

We thank the Reviewer for the comment. He/she may have missed it, but it is defined in the Introduction section (page 4, paragraph 1) where it first appears.

Reviewer #2's comment #7:

There are some sentences that could use rewarding, e.g., “were straightforward and not very complicated”, “wavelength-switched pulse source generates Stokes pulses whose wavelength is switched”, etc

Authors' response:

We thank the Reviewer for the suggestion. We have revised the sentences to make the text easier to understand as follows: “not very complicated (e.g., not as complicated as the morphology of platelet aggregates which is highly diverse and may contain white blood cells as shown in our previous work)” (page 9, paragraph 3); “wavelength-switched pulse source generates Stokes pulses in four colors” (page 5, paragraph 2).

Reviewer #2's comment #8:

This statement in the abstract seems a bit too strong “imaging-based cell picking has revolutionized single-cell biology”. Would the authors say that image-based flow cytometry is now the norm? Or is it extremely promising for now?

Authors' response:

We thank the Reviewer for the comment. While we think it's promising, we agree that the statement is a bit too strong. Therefore, we have softened the statement in both the Abstract and Introduction to address his/her

comment as follows: “The advent of image-activated cell sorting and imaging-based cell picking has advanced our knowledge and exploitation of biological systems in the last decade”.

Reviewer #2’s comment #9:

In sum, this is an excellent piece of work, with complex imaging, computational, and analysis components, demonstrated carefully on various cell types.

Authors’ response:

We thank the Reviewer for recognizing our work and giving us the positive comment.

TO REVIEWER #3

Reviewer #3’s comment #1:

Key results: In this manuscript, the authors demonstrated “Raman” image activated cell sorting by integration of ultrafast multicolour stimulated Raman Scattering microscopy, digital image processing, and a microfluidic cell sorter. The system has been applied to sort various types of cells with different treatments, illustrating a maximum throughput of ~100 events per second.

Authors’ response:

We thank the Reviewer for recognizing our work.

Reviewer #3’s comment #2:

Originality and significance: The concept of this work is not new. It is the combination of the previous work from the same group, i.e. image-activated cell sorting (Nitta, Cell 2018) and label-free chemical imaging flow cytometry by high-speed multicolour stimulated Raman Scattering (Suzuki, PANS 2019). Most of the technical platforms (including microfluidic sorter) in this work have been illustrated in their previous publications.

As the authors claimed, there is some technical development in terms of imaging acquisition scheme (Ling 169) and image construction architecture (line 198). The current implementation uses a parallel detection system and constructs images from 24-parallel 4 -colour SRS signals. These perhaps have shortened on-line processing time, which is necessary for high throughput sorting. Clearly, the group are experts in stimulated Raman Scattering (SRS) and imaging-activated cell sorting. This work shows the capability of SRS imaging for

high throughput cell sorting.

Authors' response:

We thank the Reviewer for recognizing the technical novelty of our work. The very basic concept of the work may not be new, but we would like to stress that the actual work was far from simple and required a careful design of both the major subsystems as well as a dedicated integration of them into the functional system.

Reviewer #3's comment #3:

However, the title of "Raman image activated cell sorting" is misleading. Generally, Raman often refers to spontaneous Raman spectroscopy. However, the work is based on Stimulated Raman Scattering. This is not mentioned anywhere in the title and abstract.

Authors' response:

We thank the Reviewer for the comment. The original abstract had "via coherent Raman scattering", but perhaps this was not clear. To address this point, we have made modifications to the abstract to clarify the contribution of stimulated Raman scattering to the RIACS as follows: "The advent of image-activated cell sorting and imaging-based cell picking has advanced our knowledge and exploitation of biological systems in the last decade. Unfortunately, they generally rely on fluorescent labeling for cellular phenotyping, an indirect measure of the molecular landscape in the cell, which has critical limitations. Here we demonstrate "Raman" image-activated cell sorting by directly probing chemically specific intracellular molecular vibrations via ultrafast multicolor stimulated Raman scattering (SRS) microscopy for cellular phenotyping. Specifically, the technology enables real-time SRS-image-based sorting of single live cells with an unprecedented throughput of up to ~100 events per second without the need for fluorescent labeling. To show the broad utility of the technology, we show its applicability to diverse cell types and sizes. The technology is highly versatile and holds promise for numerous applications that are previously difficult or undesirable with fluorescence-based technologies." As for the title, we would like to keep it short since the target reader of this paper is researchers in the biomedical community that do not care much about the difference between spontaneous Raman and coherent Raman in comparison to the difference between Raman and fluorescence, but do care about what the technology can do.

Reviewer #3's comment #4:

The authors seem to overstate the advantage of their approach by saying "our Raman image-activated cell sorter (RIACS) offers not only ~100 times higher throughput, but also ~600 times richer information content

by providing Raman images ..., making the RIACS ~60,000 times more powerful in terms of both population-level and single-cell-level information contents than the state-of-the-art technology". This "60,000 times more powerful.." doesn't make any sense. Every approach has pros and cons. Spontaneous Raman spectroscopy is slow, however, it offers fine spectral resolution and is well-established in biological communities. RIACS only uses 4-colour SRS imaging, which is essential for its fast spectra acquisition that comes at the expense of spectral resolution. In addition, Figures 3, 4 and supplementary figure 9 show the spectral features are sufficient to differentiate cells, and the spatial information contained in the images does not seem to add significantly to this.

Authors' response:

We thank the Reviewer for the comment. We agree with him/her that perhaps the original statement was somewhat misleading as it could be interpreted unfair to researchers who work on spontaneous Raman spectroscopy. To address his/her comment and clarify our claims, we have revised the statement as (page 4, paragraph 1): "As a result, our Raman image-activated cell sorter (RIACS) offers ~100 times higher sorting throughput as well as spatial resolution by providing Raman images in comparison to (non-imaging) Raman-activated cell sorting that provides only one-dimensional (1D) Raman signal intensities with a moderate throughput of ~1 eps." As for the second point, the SRS images do provide meaningful information to differentiate cells. For example, as shown in Figure 4b, the lipid amount alone (i.e., the product of the lipid density in the x axis and the cell area) is not sufficient for accurately differentiating adipocyte-like cells of interest; the intracellular spatial distribution of lipid droplets that can only be provided by imaging is an important factor for studying obesity. Another example for which imaging is useful is the differentiation and isolation of highly productive microalgal cells that produce a large amount of lipids without increasing the cell size. Non-imaging measurements cannot discern these cells only by their Raman/fluorescence intensity values.

Reviewer #3's comment #5:

Although the manuscript is well written, there are still some issues as detailed below:

1) The authors didn't clarify where the claimed ~100 events per second comes from. The calculation of the evident rates e_i in the "sorting throughput" section is confusing. Do the histograms of event rates in Figure 2d suggest the event rates vary in a single experiment?

Authors' response:

We thank the Reviewer for the comment. As he/she points out, we used the histograms in Figure 2d to calculate the throughput values. In ideal experiments, these histograms should have the shape of the Poisson distribution in steady state as described in the subsection "Relation between throughput and purity". However, in practice, the event rate is zero at the beginning of a sorting run, gradually rises as the injected cells reach the

microfluidic chip, and reaches steady state (i.e., Poisson distribution) during which the event rate is subject to the pre-flushing and sorting procedures of the fluidic system. Therefore, the left tail of each histogram indicates the process of the beginning of the sorting run while the central part of the histogram is the Poisson distribution of the events. Following the conventions of the field of flow cytometry, as described in the subsection “Sorting throughput”, we used the mean value of the Poisson distribution (most frequent value of the event rate) as the nominal throughput, that is, 85.2 eps and 50.2 eps from Figure 2d.

Reviewer #3’s comment #6:

2) *In the “Sorting experiment” section, the authors mention that the diameter of 1 μ m particles sample stream is 20 μ m. This indicates poor focusing for small particles. The authors should discuss the limitations of the system for sorting small objectives, since the potential applications proposed in the discussion include small sized cells (e.g. synthetic-cell sorting and bacterial-cell sorting).*

Authors’ response:

We thank the Reviewer for the comment. As he/she points out, the width of the 1- μ m particle stream is 20 μ m, but it does not mean poor focusing. In fact, since the SRS microscope’s field of view is 33 μ m (length along the flow direction) \times 24 μ m (width perpendicular to the flow direction) as stated in the subsection “Signal processing in the real-time Raman image processor”, as long as the stream of particles or cells is within 24 μ m, the SRS microscope can capture their images regardless of their size. Furthermore, our microfluidic focuser is optimized for focusing cells whose size range is 3-20 μ m within which most cell types are. For applications that involve smaller cells (<3 μ m), we can decrease the size of the microfluidic channel and optimize its design to make the stream more tight. Also, we can implement 3D hydrodynamic focusing instead of 2D hydrodynamic focusing used in this work because the focusing performance of hydrodynamic focusing is independent of cell size. Another effective focusing technique is inertial focusing, which has been used to demonstrate focusing of bacteria with submicrometer resolution (Cruz et al., “Inertial focusing with sub-micron resolution for separation of bacteria”, *Lab on a Chip* 2019, 19, 1257-1266; Zhang et al., “Focusing of sub-micrometer particles in microfluidic devices”, *Lab on a Chip* 2020, 20, 35-53). To clarify this point as well as to discuss other potential limitations and how to overcome them, we have added the following paragraph and related references to the Discussion section in the revised manuscript (page 10, paragraph 2): “In this Article, the sorting throughput, number of colors, number of pixels, and cell size range were demonstrated to be \sim 100 eps, 4, 122 \times 24 pixels, and 3-20 μ m, respectively, but these values are not fundamental physical limits and can be improved or adjusted by employing a more advanced or different architecture for the entire RIACS system as well as its major components such as the SRS microscope and real-time Raman image processor. First, the throughput can simply be boosted by increasing the flow speed of cells, but this comes at the expense of imaging sensitivity and the number of pixels in the flow direction.

Therefore, an improved design for the SRS microscope with higher sensitivity and more pixels is necessary for improving the throughput although a good balance between the sensitivity, number of pixels, and throughput needs to be taken into account, depending on the application. Second, the number of colors can be increased by increasing the number of wavelength channels in the pulse-pair-resolved wavelength-switchable laser (Supplementary Figure 3b). Since the linewidth of each wavelength channel is $<10\text{ cm}^{-1}$ and the bandwidth of the SRS microscope is about 300 cm^{-1} , the number of colors can, in principle, be boosted to ~ 30 , assuming that there is no spectral overlap between the consecutive channels. However, it is not meaningful to have more than several colors in this high-frequency spectral region since only several independent components can be resolved in the region even if the full spectral information is obtained. Third, the number of pixels can be increased by adding photodetection circuits, lock-in amplifiers, and digitizers (Figure 1, Supplementary Figure 4). Finally, while in this Article the microfluidic chip of the RIACS (i.e., microfluidic channel, cell focuser, cell sorter) were optimized for handling samples containing cells whose size range is $3\text{-}20\text{ }\mu\text{m}$ in diameter where most cell types are, a different design for the microfluidic chip is required to handle samples containing smaller cells ($<3\text{ }\mu\text{m}$), larger cells ($>20\text{ }\mu\text{m}$), or mixtures of both. For example, for applications that involve smaller cells ($<3\text{ }\mu\text{m}$) such as bacteria, other focusing techniques such as 3D hydrodynamic focusing with a narrower microchannel (instead of 2D hydrodynamic focusing used in this work), inertial focusing, deterministic lateral displacement, hydrophoretic focusing, and viscoelastic focusing^{48,49} can be implemented to focus them more tightly. In summary, a careful design, adaptation, adjustment, and optimization of the system are required, depending on the application, since these parameters are not independent, but are interrelated, whereas a more sensitive SRS microscope is expected to boost the overall system performance including the throughput, number of colors, number of pixels, and cell size range.”

Reviewer #3's comment #7:

3) If the authors sort particles one by one, they should provide evidence.

Authors' response:

We thank the Reviewer for the suggestion. To address it, we have added to the revised manuscript a supplementary video (Supplementary Movie 1) that shows real-time sorting of polymer particles by the RIACS at about 50 eps.

Reviewer #3's comment #8:

4) For the cell sorting experiments, no information is given about statistics. It is not clear whether the purity and yield data are average values or from a single experiment.

Authors' response:

We thank the Reviewer for the comment. Each sorting data was obtained from a single experiment. This is because the throughput value varies in each experiment (which is influenced by variations in the concentration of particles or cells due to their gravitational sedimentation in the sample tube during the experiment) and is in the trade-off relation with the purity as described in the subsection “Relation between throughput and purity” in the manuscript. There exists a similar relation between the throughput and yield. Therefore, unless the throughput values are identical in multiple experiments, it is meaningless to directly compare the purity and yield values in the experiments (and hence calculate their statistics) because these parameters are not independent, but interrelated. This condition is not unique to the RIACS, but is applicable to all flow cytometers as well as all types of particles and cells.

Reviewer #3's comment #9:

5) In figure 2b, the scale bar is missing.

Authors' response:

We thank the Reviewer for pointing it out. We have added the scale bar to the figure in the revised manuscript.

Reviewers' Comments:

Reviewer #2:

Remarks to the Author:

The authors responded to all my comments thoroughly. I look forward to seeing this paper in print.

Reviewer #3:

Remarks to the Author:

The authors have addressed the comments. I support it to be published as it is.